# Acetyl-CoA-Carboxylase 1-mediated de novo fatty acid synthesis sustains Lgr5+ intestinal stem cell function

Shuting Li[1], Chia-Wen Lu[1], Elia C. Diem [1], Wang Li[2], Melanie Guderian[1], Marc Lindenberg [1], Friederike Kruse[3], Manuela Buettner [4], Stefan Floess [3], Markus R. Winny[5], Robert Geffers [6], Hans-Hermann Richnow [2], Wolf-Rainer Abraham[7,8], Guntram A. Grassl[1,9] & Matthias Lochner [1,10 ✉]

Basic processes of the fatty acid metabolism have an important impact on the function of intestinal epithelial cells (IEC). However, while the role of cellular fatty acid oxidation is well appreciated, it is not clear how de novo fatty acid synthesis (FAS) influences the biology of IECs. We report here that interfering with de novo FAS by deletion of the enzyme Acetyl-CoA-Carboxylase (ACC)1 in IECs results in the loss of epithelial crypt structures and a specific decline in Lgr5+ intestinal epithelial stem cells (ISC). Mechanistically, ACC1-mediated de novo FAS supports the formation of intestinal organoids and the differentiation of complex crypt structures by sustaining the nuclear accumulation of PPARδ/β-catenin in ISCs. The dependency of ISCs on cellular de novo FAS is tuned by the availability of environmental lipids, as an excess delivery of external fatty acids is sufficient to rescue the defect in crypt formation. Finally, inhibition of ACC1 reduces the formation of tumors in colitis-associated colon cancer, together highlighting the importance of cellular lipogenesis for sustaining ISC function and providing a potential perspective to colon cancer therapy.

[1] Institute of Medical Microbiology and Hospital Epidemiology, Hannover Medical School, Hannover, Germany. [2] Department Isotope Biogeochemistry, Helmholtz Centre for Environmental Research – UFZ, Leipzig, Germany. [3] Experimental Immunology, Helmholtz Centre for Infection Research, Braunschweig, Germany. [4] Institute for Laboratory Animal Science, Hannover Medical School, Hannover, Germany. [5] Department of General, Visceral and Transplant Surgery, Hannover Medical School, Hannover, Germany. [6] Genome Analytics, Helmholtz Centre for Infection Research, Braunschweig, Germany. [7] Chemical Microbiology, Helmholtz Centre for Infection Research, Braunschweig, Germany. [8] Microbial Drugs, Helmholtz Centre for Infection Research, Braunschweig, Germany. [9] German Center for Infection Research (DZIF) partner site Hannover-Braunschweig, Braunschweig, Germany. [10] Institute of Infection Immunology, TWINCORE, Center for Experimental and Clinical Infection Research; a joint venture between the Hannover Medical School and the Helmholtz Centre for Infection Research, Hannover, Germany. ✉email: lochner.matthias@mh-hannover.de

The intestinal epithelium constitutes a physical barrier that segregates the host from the luminal environment. It is also responsible for the uptake of nutrients and contributes to the tuning of the intestinal immune system. The multitasking-ability of the epithelium is facilitated by the different sub-populations of intestinal epithelial cells (IEC). Research during the recent years provided evidence that diet- and microbiota-derived metabolites, but also the intracellular metabolism of IECs itself, critically influence the function of the epithelium.

Products of microbial fermentation of fibers, such as the short-chain fatty acid butyrate, have been demonstrated to serve as a major source of energy for IECs in the colon. Butyrate is converted to Acetyl-CoA and further metabolized by oxidative phosphorylation (OXPHOS) in the mitochondria to gain a maximal yield of energy. Oxygen consumption by this process results in deoxygenation of the epithelium, which enables the flourishing of a beneficial anaerobic microbiota and the stabilization of the epithelial barrier[1,2]. Intestinal pathogens like *C. rodentium* can subvert this beneficial metabolic circuit by shifting the metabolism of IECs towards a glycolytic profile, which leads to re-oxygenation of the epithelium and the establishment of a microenvironment that favors the outgrowth of the pathogen[3]. Importantly, under physiological conditions the consumption of butyrate by enterocytes protects a set of proliferating cells at the bottom of the epithelial crypt structures from the toxic effect of this metabolite[1]. Such crypt base columnar cells have been identified as multipotent, self-renewing Lgr5[+] intestinal epithelial stem cells (ISC), which constantly generate transit amplifying progenitors that give rise to all other cell types of the epithelium[4]. In the small intestine, Lgr5[+] cells are interspersed with paneth cells, which together with the ISCs constitute the stem cell niche. Paneth cells, beside their ability to produce antimicrobials, provide ISCs with important factors (e.g. Wnt, Delta-like 1 and epidermal growth factor) to support their stem cell function[5]. Recent research also indicated that paneth cells and ISCs comprise a metabolic entity, in a way that paneth cells employ a glycolytic metabolism to produce lactate, which is then further metabolized by ISCs via mitochondrial OXPHOS[6]. The stem cell niche also integrates dietary signals into functional outcomes. As such, calorie restriction augmented the capacity of paneth cells to boost ISC function in a process involving the key metabolic regulator mTOR[7]. Moreover, feeding mice with either a ketone rich or high glucose diet directly affected ISC maintenance and function through the generation of ketone bodies in ISCs[8]. Together, these findings indicate that different cell types of the intestinal epithelium conduct specific metabolic programs, which can be adjusted by both intrinsic and extrinsic signals.

In this regard, evidence is emerging that processes involved in the metabolism of fatty acids have a critical impact on the regulation of ISC maintenance and function. It has been demonstrated that reducing the rate of intracellular fatty acid oxidation (FAO) in epithelial cells, for example by deleting the fatty acid binding transcription factor HNF4 or the enzyme Cpt1a, which shuttles long- and medium-chain fatty acids into mitochondria for FAO, decreased ISC numbers and function[9,10]. Likewise, excess external delivery of lipids or cholesterol through the diet fostered ISC renewal and progenitor cell proliferation, but as a consequence also rendered mice more susceptible towards intestinal tumor formation[11,12]. Besides the uptake of lipids from the environment, cells can also produce their own long chain fatty acids from metabolic precursors, namely Acetyl-CoA, in a process termed de novo fatty acid synthesis (FAS). Interestingly, disruption of FAS can affect the function of cells in a cell type-specific manner, despite the presence of physiological amounts of lipids in the environment. In that respect, work from our lab demonstrated recently that inhibition of the rate limiting enzyme for de

novo FAS, Acetyl-CoA carboxylase (ACC) 1, which catalyzes, the ATP-dependent carboxylation of Acetyl-CoA to Malonyl-CoA, affects the differentiation of specific T cell subsets and interferes with the function of innate lymphoid cells in the intestine[13,14]. In addition, it has been suggested that de novo FAS is critical to maintain embryonic stem cell pluripotency and to promote induced pluripotent stem cell formation[15,16]. Yet, despite the appreciated role of de novo FAS for cellular differentiation and function, the significance of this process in IECs is not clear so far.

In this study, we demonstrate that intestinal epithelium-restricted deletion of ACC1 results in damaged crypt formation in the small intestine and the colon of mice. Inhibition of de novo FAS interferes with both the ISC-driven growth of intestinal organoids as well as with the development of crypt domains, indicating a specific function for de novo FAS in Lgr5[+] ISCs. We show that disrupting de novo FAS affects organoid development by diminishing the accumulation of intracellular lipids and restricting the activation of the PPARδ/β-catenin signaling pathway. External supplementation of an excess amount of fatty acids is sufficient to rescue the loss of de novo FAS in vitro and in vivo, demonstrating that the fat content of the diet can regulate the dependency of ISCs on their own ability to synthesize fatty acids. Importantly, functional inhibition of ACC1 in the intestinal epithelium limits the formation of tumors in an inflammation-driven model of colon cancer, providing insights into the role of de novo FAS in colon cancer development and therapy.

## Results

**Epithelial ACC1 deletion compromises intestinal crypts.** To determine the role of ACC1-mediated de novo FAS in the intestinal epithelium, we generated IEC-specific ACC1 knockout mice by crossing ACC1[lox/lox] mice with the tamoxifen-inducible Villin-Cre[ERT2] strain (hereafter referred to as ACC1[Δ/ΔIEC] mice). To confirm IEC-specific deletion, PCR was performed with primers specifically recognizing the deletion of exon 22 of the *Acaca* gene (encoding for ACC1)[17]. Since villin-driven cre recombinase expression in renal epithelium has previously been reported in Villin-Cre[ERT2] mice[18], PCR was performed using kidney tissue and the spleen as additional controls. As shown in Supplementary Fig. 1a, ACC1 deletion was only found in IECs from small intestine and colon, but not in kidney or spleen.

To assess a potential effect on the physiology of the intestine, we analyzed mice 2 weeks upon tamoxifen-mediated ACC1 deletion. qPCR analysis confirmed the efficient deletion (>90%) of functional ACC1 in IECs derived from both the small intestine and the colon at this time point (Fig. 1a). Importantly, histopathological assessment of the small intestine revealed clear signs of damaged crypt structures in ACC1[Δ/ΔIEC] mice, with increasing severity from the proximal towards the distal parts, resulting in an almost complete loss of crypt structures in the ileum (Fig. 1b). Also in the distal part of the colon, the epithelial architecture was severely affected, as shown by deformed crypt structures and the presence of crypt abscesses (Fig. 1c). This was accompanied by a pronounced cellular infiltration into the small intestinal lamina propria of ACC1[Δ/ΔIEC] mice (Fig. 1b). Of note, earlier inspection of the phenotype on day 6 after the first tamoxifen treatment did not reveal histopathological changes, indicating that manifestation of the phenotypic changes is rather slowly progressing (Supplementary Fig. 1b). Closer investigation of the cellular content of the lamina propria by flow cytometry showed increased frequencies of innate leukocytes such as dendritic cells, neutrophils and macrophages in the small intestine, but not in the colon of ACC1[Δ/ΔIEC] mice (Fig. 1d and Supplementary Fig. 1c). Despite the epithelial pathology and the

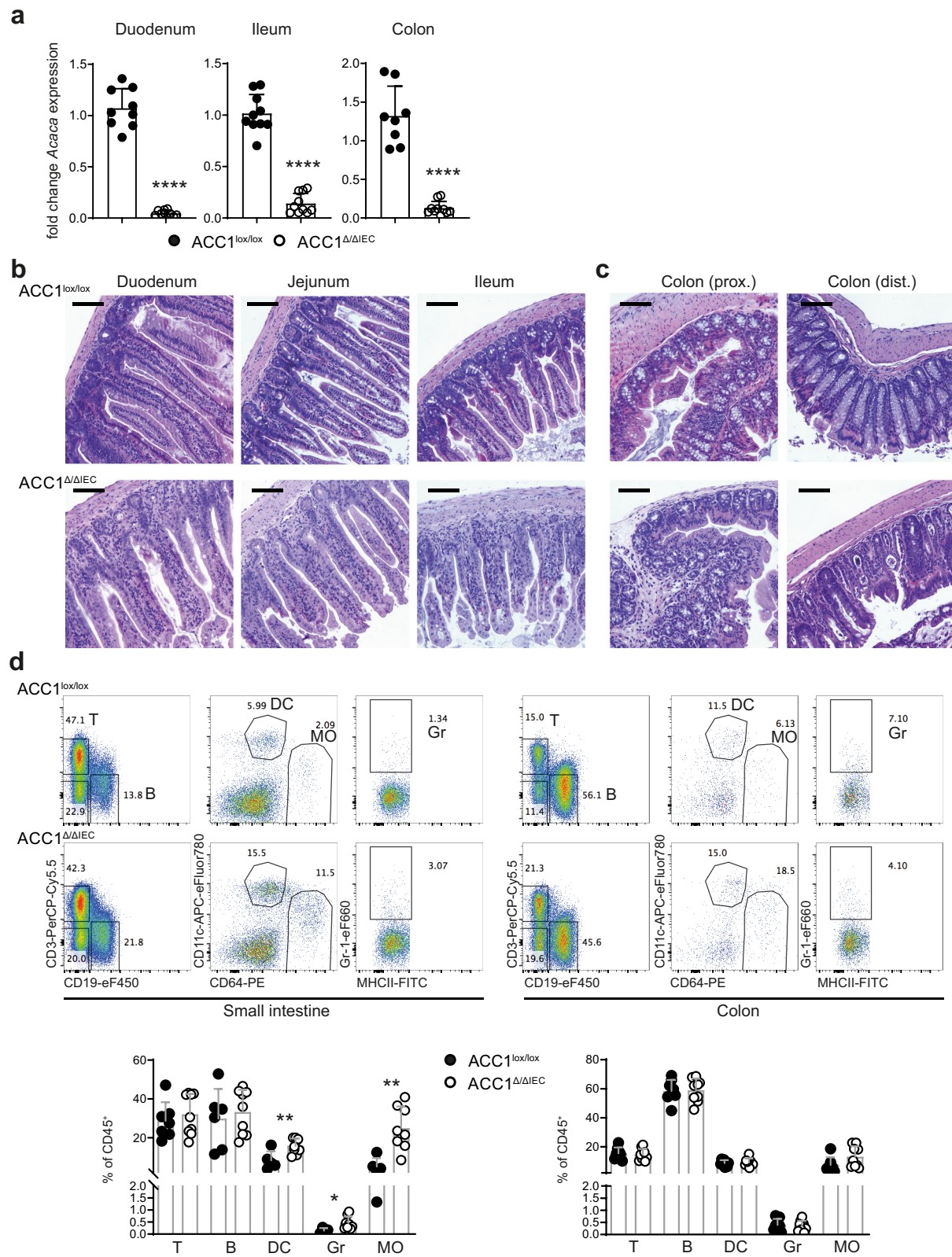

signs of local inflammation, the mice did not show overt signs of disease or weight loss (Supplemental Fig. 1d), presumably due to the less affected proximal parts of the intestine. Analysis at 2 months after tamoxifen treatment confirmed that the histopathological changes were preserved in the lower parts of both the small intestine and the colon (Supplementary Fig. 1e). Together, our findings demonstrate a critical function for ACC1

in the maintenance of a normal epithelial crypt architecture, indicating an important function for de novo FAS in IEC biology.

**Epithelial loss of ACC1 affects Lgr5$^+$ ISCs.** Our finding of missing or damaged crypts in ACC1$^{\Delta/\Delta IEC}$ mice suggested that ACC1-deletion might preferentially interfere with the maintenance or function of crypt residing cell types, in particular the stem cell

**Fig. 1 Epithelium-specific ACC1 deletion affects crypt architecture in the small intestine and the colon.** ACC1$^{\Delta/\Delta IEC}$ and ACC1$^{lox/lox}$ control mice were treated with tamoxifen (i.p.) for 5 consecutive days and analyzed on day 9 upon the last tamoxifen injection. **a** RNA was extracted from duodenum, ileum and colon epithelial cells and qPCR was performed to determine ACC1 deletion efficiency. Data was pooled from 2 independent experiments with a total of $n = 9$ and 10 mice per group. **b** Representative H&E stainings of duodenum, jejunum and ileum and **c** of the proximal (prox.) and distal (dist.) part of colon from ACC1$^{lox/lox}$ control (upper panel) or ACC1$^{\Delta/\Delta IEC}$ mice (lower panel) from >3 independent experiments with $n \geq 3$ mice per group are shown. Scale bars represent 100 μm. **d** Leukocytes were isolated from the lamina propria of the colon and small intestine and analyzed by flow cytometry. Representative flow cytometry plots from ACC1$^{lox/lox}$ control and ACC1$^{\Delta/\Delta IEC}$ mice are shown (upper panel). Graphs show frequency (%) of live CD45$^{+}$ T cells (T), B cells (B), dendritic cells (DC), neutrophilic granulocytes (Gr) and macrophages (MO), Data was pooled from 2 independent experiments with a total of $n = 8$ and 9 mice per group. Statistical significance was analyzed using unpaired two-tailed Student's $t$-test (**a**, **d**) with $*p < 0.05$, $**p < 0.01$, $***p < 0.001$. $****p < 0.0001$. Exact $p$ values provided as Source Data. Bar graphs represent mean and error bars indicate SD. Source data are provided as a Source Data file.

compartment. To further investigate this hypothesis, we first assessed the expression of the ISC-associated marker Lgr5 in IECs. As shown in Fig. 2a, epithelium-specific ACC1 deletion resulted in significant downregulation of *Lgr5*-expression in IECs. To further confirm the specific effect of IEC-driven deletion of ACC1 on the maintenance of Lgr5$^{+}$ ISCs, we directly assessed the presence of these cells in histology. To this end, we used an antibody directed against Olfm4, which is another described marker for Lgr5$^{+}$ ISCs in the small intestine[19]. We indeed found that in the small intestine of ACC1$^{\Delta/\Delta IEC}$ mice, the Olfm4$^{+}$ cell population was reduced (Fig. 2b), which supported the idea that ACC1 deletion resulted in the loss of Lgr5$^{+}$ ISCs. At the same time, we noted that ACC1 deletion did not appear to have a strong effect on the expression of markers associated with other epithelial cell types, such as *Muc2* (goblet cells), *Lyz1* (paneth cells) or *Chgb* (enteroendocrine cells) (Fig. 2c). Direct immunohistological assessment of paneth cells showed a slight increase in the numbers of paneth cells in the small intestine of ACC1$^{\Delta/\Delta IEC}$ mice, while periodic acid-schiff (PAS)-staining of goblet cells did not show notable changes in the number of these cells (Fig. 2d). Besides the highly proliferative Lgr5$^{+}$ base columnar ISCs, a more quiescent subpopulation of stem cells can be found residing around position +4 in the paneth cell zone, where they express markers such as *Tert*[20]. However, our results indicated that in contrast to Lgr5, the expression of *Tert* remained largely unaffected in the intestinal epithelium of ACC1$^{\Delta/\Delta IEC}$ mice (Fig. 2c). To further study the cell-autonomous role of ACC1 in Lgr5$^{+}$ ISCs, we isolated Lgr5$^{+}$ cells from the small intestine of Lgr5-EGFP-IRES-Cre$^{ERT2}$ reporter mice[4] and first confirmed ACC1 expression in Lgr5-EGFP$^{high}$ ISCs (Supplementary Fig. 2a). Next, we bred Lgr5-EGFP-IRES-Cre$^{ERT2}$ to ACC1$^{lox/lox}$ mice, in order to allow for specific inactivation of ACC1 in Lgr5$^{+}$ ISCs (ACC1$^{\Delta/\Delta Lgr5}$ mice). Although Lgr5-specific deletion of ACC1 did not result in different frequencies of EGFP-expressing ISCs (Lgr5-EGFP$^{high}$) and progenitor cells (Lgr5-EGFP$^{int}$) in the proximal part of the small intestine, frequencies of ISCs and progenitors were significantly reduced in the distal part of the small intestine and the colon of ACC1$^{\Delta/\Delta Lgr5}$ mice (Fig. 2e and Supplementary Fig. 2b). Together, these results suggest that Lgr5$^{+}$ ISC are specifically compromised by ACC1 deletion, while most other epithelial cell types seem to be less affected.

**ACC1 is required for growth and differentiation of intestinal organoids.** In order to further elucidate how ACC1 deletion interferes with epithelial growth and function, we tested the potential of isolated crypts to form clonal, multipotent organoid bodies in the absence of ACC1 in vitro[21]. To this end, we isolated intestinal crypts form ACC1$^{\Delta/\Delta IEC}$ and ACC1$^{lox/lox}$ control mice, treated them for 24 h with the active metabolite of tamoxifen (4-hydroxytamoxifen, 4-OHT) to delete ACC1, and assessed their ability to develop complex crypt structures in 3D organoid cultures. As expected, organoids from ACC1$^{lox/lox}$ control mice formed into spheres on day 1 and developed complex structures

with numerous crypt domains on day 4 of the culture (Fig. 3a, red arrows indicate crypt domains). In contrast, upon in vitro deletion of ACC1 (Fig. 3b), organoids did not further develop into complex structures and lacked the typical crypt domains on day 4 (Fig. 3a, c), which was accompanied by a progressive loss of *Lgr5* expression in ACC1-deficient organoids (Fig. 3d). RNA-based next-generation sequencing (RNA-seq) analysis of organoids at day 4 of culture confirmed a markedly lower expression of ISC-associated genes *Lgr5*, *Olfm4*, *Smoc2*, *Msi1 Ascl2*, as well as *Tert* in ACC1-deficient organoids (Fig. 3e). In contrast, expression of markers of the secretory lineage of epithelial cell types (*Atoh1*, *Muc2*, *Lyz1*, *Reg3g*, *Chgb*) or absorptive enterocytes (*Slc5a11*, *Alpi*) was rather enhanced (Fig. 3f). The increase of secretory epithelial cells upon loss of ACC1 was confirmed by directly staining for Muc2 in organoids (Supplementary Fig. 3a) together indicating a change in the ratio between ISCs and differentiated cell types under in vitro conditions. Gene set enrichment analysis (GSEA) indicated that genes associated with DNA replication, cell cycle progression and chromosome segregation were all significantly downregulated in ACC1-deficient organoids (Fig. 3g and Supplementary Fig. 3b, c), suggesting that loss of ACC1-mediated de novo FAS specifically affects the proliferation and self-renewing capacity of ISCs. In line with these results, we observed increased frequencies of dying cells in ACC1-deficient organoids (Supplementary Fig. 3d) and a severely impaired capacity to form secondary organoids upon sub-cloning (Fig. 3h). Finally, we confirmed the reduced organoid-forming potential of crypts isolated ex vivo from tamoxifen-treated ACC1$^{\Delta/\Delta IEC}$ mice (Fig. 3i).

To further dissect the impact of de novo FAS during initial growth into spherical organoids and the development of complex crypt domain containing structures, we sorted Lgr5$^{+}$ ISCs from the small intestine of Lgr5-EGFP-IRES-Cre$^{ERT2}$ mice and cultured them in the presence of the ACC-specific inhibitor Soraphen A (SorA) to inhibit ACC1-mediated de novo FAS[13]. We used a conditioned growth medium, which promotes the growth of spherical organoids that lack differentiated cell types[6]. After 7 days in the conditioned growth medium, isolated Lgr5$^{+}$ cells expanded and self-organized into spherical organoids. However, SorA treated spheroids were much smaller in size as compared to the spheres grown with vehicle control, indicating that ACC1-inhibition interferes with the homogenous ISC proliferation-driven initial growth of spherical organoids (Fig. 3k, upper panel). To test the effect of ACC1-inihibtion specifically during the differentiation phase, organoids grown for 7 days in conditioned growth medium were changed to a differentiation medium that efficiently supports the formation of crypt domains. As shown in Fig. 3k (lower panel), vehicle-treated organoids rapidly developed multiple crypts within 2 days upon change of the culture medium, whereas addition of SorA to the differentiation medium effectively prevented crypt domain formation. Together, these results demonstrate that inhibition of

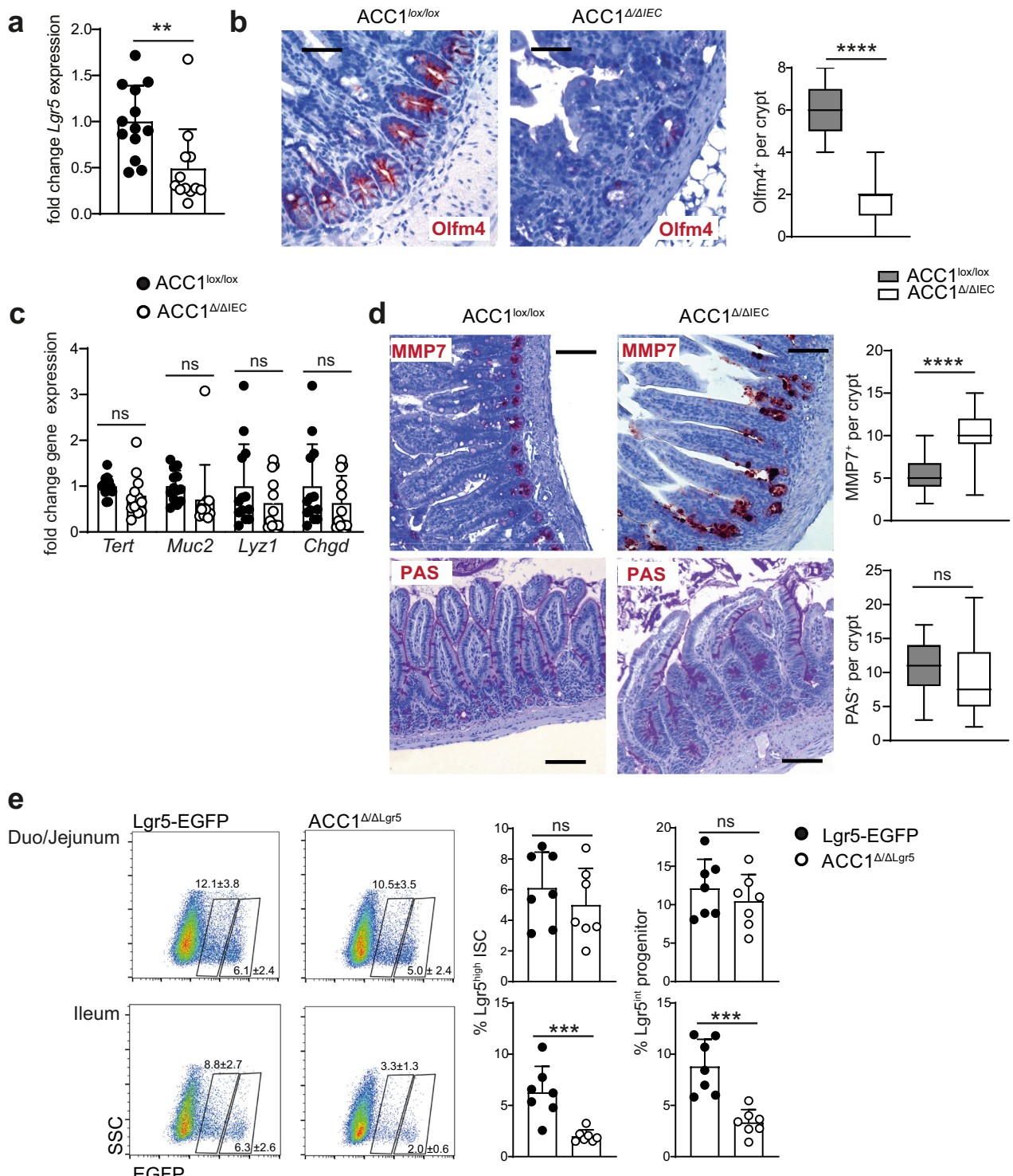

ACC1 specifically affects Lgr5+ stem cell function during both, initial organoid growth as well as the differentiation/crypt budding phase. Of note, we confirmed that similar to what we observed in mouse-derived organoids, inhibition of ACC1 with SorA prevented the formation of complex structures also in organoids grown from isolated human intestinal epithelial crypts (Fig. 3l).

Paneth cells can support ISC function by the production of Wnt ligands, and thus enhance the growth of organoids when co-cultured together with sorted Lgr5+ ISCs[5]. Importantly, Wnt-ligands can be modified post-translationally by lipids (e.g. by palmitoylation), which has been suggested to be an important feature for Wnt ligand trafficking[22]. In our next experiments, we wanted to test whether ACC1 inhibition may affect organoid development by interfering with the lipidation of Wnt ligands produced by paneth cells. Therefore, we sorted paneth cells from ACC1$^{\Delta/\Delta IEC}$ mice and co-cultured them with (ACC1-sufficient) Lgr5-EGFP$^{high}$ cells isolated from Lgr5-EGFP-IRES-cre$^{ERT2}$ mice. As expected, organoids from Lgr5+ ISCs co-cultured with paneth cells grow faster than in the absence of those cells

**Fig. 2 IEC-specific ACC1-inactivation results in a specific loss of Lgr5+ ISCs.** ACC1$^{\Delta/\Delta IEC}$ and ACC1$^{lox/lox}$ control mice were analyzed on day 9 after a 5 day period of tamoxifen treatment. **a** Expression of *Lgr5* gene in IECs isolated from the small intestine. Data was pooled from 3 independent experiments with a total of $n = 12$ and 13 mice per group. **b** Immunohistochemical staining of the small intestine (ileum) with anti-Olfm4 antibody. For quantification >10 crypts were examined per mouse. Data is shown from one representative out of 3 independent experiments with $n = 5$ mice per group, bar = 50 μm. **c** Expression of *Tert*, *Muc2*, *Lyz1* and *Chgb* in IECs isolated from the small intestine. Data was pooled from 3 independent experiments with a total of $n = 12$ and 13 mice per group. **d** Immunohistochemical staining of small intestinal paneth cells in the ileum of ACC1$^{\Delta/\Delta IEC}$ and ACC1$^{lox/lox}$ control mice with anti-MMP-7 antibody (upper panel) and PAS staining of neutral and acidic mucins (in pink), representing goblet cells (lower panel), bar = 200 μm. For quantification >10 crypts were examined per mouse. Data was pooled from 2 independent experiments with a total of $n = 3$ and 4 mice per group. **e** Frequency of DAPI⁻Epcam+Lgr5$^{high}$ ISCs and DAPI⁻Epcam+Lgr5$^{int}$ progenitor cells within total DAPI⁻Epcam+ epithelial cells isolated from the duodenum/jejunum or the ileum of Lgr5-EGFP-IRES-cre$^{ERT2}$ (Lgr5-EGFP) control and ACC1$^{\Delta/\Delta Lgr5}$ mice one day upon the last tamoxifen injection. Data pooled from 2 independent experiments with a total of $n = 7$ mice per group. Statistical significance was analyzed using unpaired two-tailed Student's *t*-test (**a–e**) with \*$p < 0.05$, \*\*$p < 0.01$, \*\*\*$p < 0.001$. \*\*\*\*$p < 0.0001$. Exact *p* values provided as Source Data. Bar graphs represent mean and error bars indicate SD. Boxes in boxplots denote 25th to 75th percentiles with whiskers representing min-max, and the central line the median. Source data are provided as a Source Data file.

(Fig. 3m). Yet, specific deletion of ACC1 in paneth cells by 4-OHT had no effect on the growth of the co-cultured organoids, indicating that inhibition of FAS does not affect paneth cells in their ability to support Lgr5+ ISC function.

**FAS supports ISC function by sustaining PPARδ/β-catenin.** We next sought to understand the molecular mechanisms by which the loss of ACC1 affects ISC function. Real time metabolic analysis of organoids showed that deletion of ACC1 had only minor effects on the oxygen consumption rate, and did not critically impact on mitochondrial metabolism or the ability to perform glycolysis (Supplementary Fig. 4a), indicating that inhibition of ACC1 did not result in a major break down of basic metabolic functions. Although we did not detect a major impact on the transcriptional level of FAS-associated genes in ACC1-deficient organoids (Fig. 4a), we confirmed that loss of ACC1 efficiently blocked the incorporation of ¹³C-labelled glucose into cellular palmitate and its immediate derivatives stearate and oleate (Fig. 4b). Accordingly, we observed reduced accumulation of cellular neutral lipids in organoids deleted for ACC1 (Fig. 4c). Of note, the addition of acetate, the precursor of the ACC1 substrate acetyl-CoA, did not rescue the phenotype of ACC1-deficient organoids (Supplementary Fig. 4c), excluding compensatory mechanisms and in summary proving the efficient shut down of de novo FAS upon ACC1 deletion.

It is well known that nuclear translocation of β-catenin upon binding of Wnt ligands to their receptors on ISCs is critical for maintaining stem cell function[23]. In accordance with these findings, we observed that interruption of de novo FAS in ACC1-deficient organoids resulted in decreased nuclear β-catenin protein levels (Fig. 4d). Previous studies have demonstrated sustained activity of β-catenin in organoids derived from mice fed with a high fat died (HFD), presumably as a consequence of direct protein interaction of nuclear PPARδ with β-catenin[11]. PPARδ has been described as master regulator of lipid metabolism and is the PPAR isoform that is highest expressed in intestinal crypts of mice[24,25]. In order to determine whether disruption of intracellular de novo FAS would affect the level of nuclear PPARδ, we performed western blot analysis of organoids deleted for ACC1. Indeed, our results revealed an increase of nuclear PPARδ protein levels in cultures of control organoids, whereas in the absence of ACC1, the amount of nuclear PPARδ protein decreased over time (Fig. 4c). To directly test a possible role of PPARδ in vivo, we treated mice with the pharmaceutical PPARδ agonist GW501516. As shown in Fig. 4e, activation of PPARδ by GW501516 strongly reduced the severe histological signs of crypt loss and destruction in the ileum of ACC1$^{\Delta/\Delta IEC}$ mice. Moreover, activation of PPARδ (but not PPARγ) increased significantly the amount of crypt domains and expression of *Lgr5*

in organoids lacking ACC1 (Fig. 4f, g and Supplementary Fig. 4c). ACC1-deficient organoids exposed to the PPARδ agonist displayed elevated nuclear protein level of both PPARδ and β-catenin (Fig. 4h), and showed increased self-renewing capacity in secondary assays (Fig. 4i). Notably, stimulation of β-catenin by addition of Wnt3a or the GSK3 Inhibitor CHIR99021 did not rescue the defect of ACC1-deficient organoids (Supplementary Fig. 4d), indicating that activation of Wnt/β-catenin alone is not sufficient to sustain ISCs in the absence of de novo FAS. Together, our results suggest that the intracellular fatty acid synthesis program can support ISCs function and that activation of PPARδ can compensate for the loss of ACC1 by maintaining β-catenin signaling.

**External lipids tune the dependency of ISCs on de novo FAS.** The C₁₆ saturated fatty acid palmitate is the end product of de novo FAS. Palmitate is a precursor for the production of cellular lipids, which can serve as important structural components in cells, but are also involved in the regulation of numerous cellular processes such as metabolism, cellular membrane motility and signaling pathways[26]. Considering the importance of de novo FAS for organoid development, we wanted to test in the next step whether external delivery of lipids may alleviate the dependency on ACC1-mediated de novo FAS. To this end, we first confirmed that both ACC1-sufficient and ACC1-deficient organoids were equally able to take up externally delivered palmitate (Supplementary Fig. 5a). As shown in Fig. 5a, b, the addition of palmitate to organoid cultures resulted in the recurrence of crypt structures and sustained *Lgr5* expression in ACC1-deficient organoids. Moreover, we observed that the amount of nuclear PPARδ and β-catenin, as well as the potential to form secondary organoids was substantially increased by palmitate addition, although not to wild type levels (Fig. 5c, d). Recent reports have highlighted the role of FAO for the maintenance and function of ISCs[9,10]. We therefore asked whether newly synthesized or externally added fatty acids may support ISC function by serving as substrates for FAO. We found that blockade of FAO using the Cpt1a inhibitor Etomoxir indeed reduced the formation of crypt domains in wildtype organoids. Yet, Cpt1a inhibition did not prevent the rescue of the ACC1-deficient phenotype by palmitate addition (Fig. 5e), indicating that the production of fatty acids to support FAO is not the main reason for the dependency of ISCs on de novo FAS. In line with these findings, we also did not observe major changes in the transcription levels of genes associated with FAO in ACC1-deficient organoids (Supplementary Fig. 5c). Finally, we aimed to evaluate whether external delivery of fatty acids would also have an effect on the in vivo phenotype. To test this, we fed ACC1$^{\Delta/\Delta IEC}$ mice for 3 weeks with a diet containing high amounts of fat (HFD). While mice that received a control

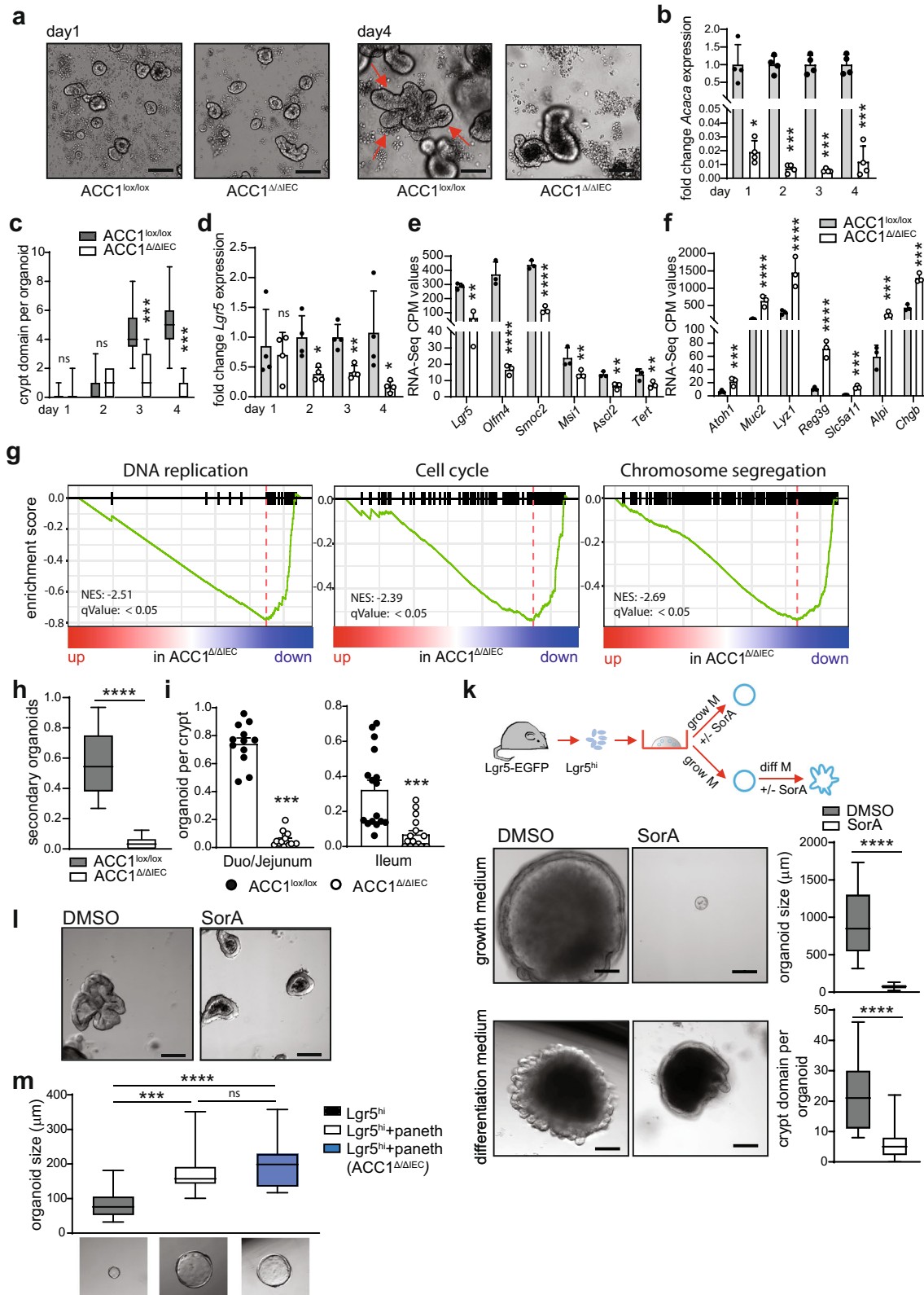

diet developed severe histological signs of crypt loss and destruction in the small intestine, the HFD fed mice showed a normal histological appearance, accompanied with a significant increase in epithelial *Lgr5* expression (Fig. 5f). Thus, the lipid content of the diet controls the dependence of the epithelium on cellular de novo FAS. In the absence of lipid-rich foods, ISCs depend on their own cellular supply of fatty acids to sustain their

function. Yet, this can be compensated by excess delivery of external fatty acids.

**Epithelial loss of ACC1 ameliorates colonic tumorigenesis.** Deregulation of Lgr5 and the Wnt signaling pathway in ISCs mark key events in the formation and progression of colorectal

**Fig. 3 ACC1 deletion restricts growth and differentiation of intestinal organoids. a–d** Organoid culture of crypts isolated from ACC1$^{\Delta/\Delta IEC}$ and ACC1$^{lox/lox}$ mice. 4-OHT was added for 24 h after plating to induce ACC1 deletion in organoids from ACC1$^{\Delta/\Delta IEC}$ mice. **a** Organoids were imaged at the indicated time points. Representative pictures are shown from one out of 4 independent experiments with similar results. Bar = 200 μm. Red arrows indicate crypt domains. **b, d** RNA was extracted from organoids at indicated time points and expression of genes was analyzed by qPCR. Data was pooled from 4 individual experiments. **c** Number of crypt domain formation in organoids at indicated time points. For quantification, more than 40 organoids per group were analyzed at each time point. Data shown for one representative out of 4 individual experiments. **e, f** Transcription levels of ISC signature genes and genes associated with secretory or absorptive IEC lineages. CPM values were derived from RNA-seq of organoids from ACC1$^{\Delta/\Delta IEC}$ and ACC1$^{lox/lox}$ mice at day 4 after 4-OHT treatment. RNA-seq was performed in triplicates from 3 independent experiments. **g** GSEA of RNA-seq for DNA replication (GO: 0006260), cell cycle (mmu04110) and chromosome segregation (GO: 0007059). **h** Number of secondary organoids per dissociated crypt-derived primary organoids. Data pooled from $N = 3$ independent experiments. **i** Number of organoids derived from ex vivo isolated crypts. Crypts were isolated from the upper (duo/jejunum) and lower part (ileum) of the small intestine and cultured in vitro. Data pooled from 3 independent experiments with $n \geq 3$ mice per group. **k** Experimental set up. SorA (200 nM) or DMSO was added during organoid culture either in growth medium (upper panel) or differentiation medium (lower panel). Representative pictures of organoids are shown from two independent experiments. Bars represent 200 μm. Quantification was done by measuring the diameter of the spheroids or counting crypt domain formation. Data was pooled from 2 independent experiments, >30 organoids were analyzed for each group. **l** Human-derived intestinal organoids were cultured for 7 days in the presence or absence of Soraphen A (1 μM). Representative pictures shown for one out of 3 independent experiments with similar results using organoids derived from colonic resections from 3 different adult male or female patients. Bar = 200 μm. **m** 500 Lgr5-EGFP$^{high}$ cells were sorted from the small intestine of Lgr5-EGFP-IRES-Cre$^{ERT2}$ mice and cultured either alone, or in the presence of 500 CD24$^{high}$SSC$^{high}$ paneth cells sorted from ACC1$^{\Delta/\Delta IEC}$ mice. Representative images of organoids are shown from two independent experiments, bar = 200 μm. For quantification, diameter of organoids was measured. Data was pooled from 2 independent experiments, $\geq$13 organoids were analyzed for each group. Statistical significance was analyzed using unpaired two-tailed Student's $t$-test (**h, i, k**), Benjamini–Hochberg procedure (**e, f**) or One-way ANOVA with Tukey's multiple comparison test (**b, c, d, m**) with $*p < 0.05$, $**p < 0.01$, $***p < 0.001$. $****p < 0.0001$. Exact $p$ values provided as Source Data. Bar graphs represent mean and error bars indicate SD. Boxes in boxplots denote 25th to 75th percentiles with whiskers representing min-max, and the central line the median. Source data are provided as a Source Data file.

cancer[27]. Furthermore, pro-obesity diets such as HFD can augment the number and proliferation of ISCs and have been associated with an increased incidence of colonic tumors[28]. Considering our findings above, demonstrating a critical role of de novo FAS for Lgr5$^+$ ISC maintenance and function, we speculated that IEC-specific ACC1 inhibition may also impact on colonic tumor formation. To test this hypothesis we employed the AOM/DSS inflammation-associated intestinal tumor model, which exhibits characteristics similar to human colitis-associated colorectal cancer, including distally-located tumors and invasive adenocarcinomas[29]. Analysis of tumor formation upon repeated AOM treatment and three rounds of DSS (Fig. 6a) revealed indeed significantly reduced numbers of tumors in the colon of ACC1$^{\Delta/\Delta IEC}$ mice (Fig. 6b, c). Yet, although less tumors formed in the absence of epithelial ACC1, their size distribution and histological appearance was similar to the tumors found in control animals (Fig. 6d). Notably, ACC1$^{\Delta/\Delta IEC}$ mice developed more severe signs of intestinal pathology, both during acute DSS colitis (Supplementary Fig. 6) and after 10 weeks of AOM/DSS treatment, as indicated by a significant decreased colon length and enhanced inflammatory tissue pathology (Fig. 6b, e). Thus, the reduction in tumor burden that we observed in ACC1$^{\Delta/\Delta IEC}$ mice cannot be explained by reduced inflammation, but rather reflects the direct consequences of ACC1-deficiency on ISC biology. In summary, our results indicate that inhibition of ACC1-mediated de novo FAS in the epithelium may directly interfere with the initial transformation and proliferation steps in ISCs. However, when initial formation of adenomas has taken place, they further progress and develop into tumors to the same extent as in colons of ACC1-sufficient control mice.

## Discussion

It has been well appreciated that the fatty acid metabolism plays an important role in the regulation of IECs, but the impact of intracellular de novo FAS has not been addressed so far. ACC1 is the rate-limiting enzyme of de novo FAS in cells. Thus, disruption of ACC1 function impairs lipid generation. Our study demonstrates that ACC1 deletion in IECs affects crypt formation in the epithelium by specifically compromising the function of Lgr5$^+$ ISCs. It is well known that the Wnt/β-catenin signaling pathway

plays an important role for the maintenance of ISCs, and our data shows that inhibition of de novo FAS in ISCs results in decreased accumulation of nuclear β-catenin. Hence, the phenotype that we describe here is in accordance with several reports showing that interference with Wnt signaling, e.g. by epithelial overexpression of the Wnt signaling pathway inhibitor Dickkopf1 or conditional deletion of the main effector molecule in the canonical Wnt signaling pathway, TCF4, strongly affects the intestinal crypt architecture[30–32].

We observed that reduction of nuclear β-catenin levels in ACC1-deficient organoids correlated with a concomitant reduction of nuclear PPARδ. Beyaz and colleagues demonstrated that feeding mice with a HFD activated the PPARδ pathway and increased the number and function of ISCs[11]. Intriguingly, long-term HFD or treatment with a PPARδ agonist enabled ISC to escape their niche dependence and equipped non-stem cells progenitors with ISC functions, indicating that PPARδ activation in ISCs might be a specific consequence of the metabolic adaptations during HFD. A recent study suggested in this context that the effect of HFD is mediated by a PPARδ-induced upregulation of FAO[33]. Our findings here support a model in which activation of PPARδ can also support ISC function in the absence of cellular de novo FAS. However, we did not observe a major impact on the expression levels of classical PPARδ downstream genes, such as Cpt1a or genes involved in FAO. In line with this, ACC1-deficient organoids did not show major defects in main metabolic functions, such as glycolysis and OXPHOS. Although our results confirm earlier reports that described a critical function of FAO in ISCs[9,10], we observed that the ACC1-deficient phenotype can be rescued in organoids when Cpt1 is blocked by Etomoxir, and therefore independently of FAO. Together, these findings indicate that metabolic adaptations, such as changes in FAO levels, do not play a major role in our system. In addition to the effect on fatty acid metabolism, PPARδ or its direct transcriptional targets have been demonstrated to influence critically on a variety of cellular functions, such as cellular differentiation, inflammation as well as stem cell maintenance[28]. In fact, direct interaction between β-catenin and PPARδ has been reported by several groups[11,34]. A seminal report by Scholtysek et al. demonstrated a direct link between PPARδ activation and stem cell function in osteoblasts[34].

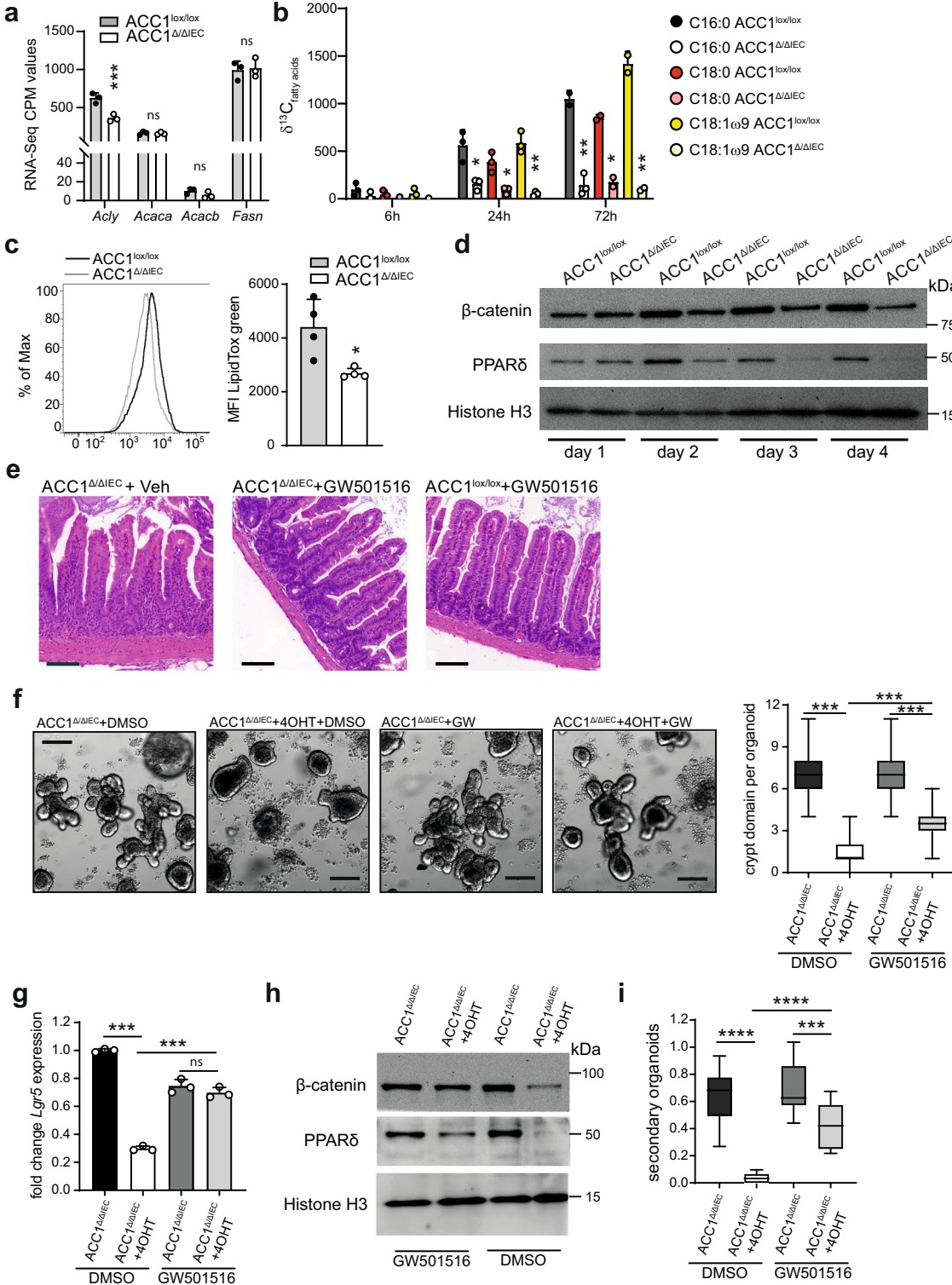

Importantly, the authors found that activation of PPARδ with GW501516 resulted in the accumulation of nuclear β-catenin in both osteoblasts and mesenchymal stem cells (which was not observed upon activation of PPARγ). Similarly, activation of PPARδ induced the Wnt-dependent mRNA expression of osteoblast marker genes. This is in accordance with our findings that PPARδ activation by GW501516 can rescue the defect of ACC1-deficiency, leading to increased nuclear β-catenin accumulation and enhanced ISC marker expression and function. Thus, activation of PPARδ may promote ISC function by directly

stabilizing β-catenin function in ISCs. Notably, organoids derived from PPARδ-deficient mice do not show overt defects under physiological conditions[33]. It is therefore possible that the role of PPARδ to support ISC function becomes functionally critical under conditions of HFD or when the internal fatty acid supply is blunted by the inhibition of de novo FAS. Whether the immediate products of de novo FAS can act as direct endogenous ligands for PPARδ is not clear. PPARδ can bind to a wide range of fatty acids and fatty acid metabolism-associated metabolites[28], and our results demonstrate that addition of palmitate, the end-product of

**Fig. 4 ACC1 deletion interferes with PPARδ/β-catenin activation in intestinal organoids. a** Transcription levels of genes associated with intracellular de novo fatty acid synthesis. CPM values were derived from RNA-seq of organoids from ACC1$^{\Delta/\Delta IEC}$ and ACC1$^{lox/lox}$ mice at day 4 after 4-OHT treatment. RNA-seq was performed in triplicates from 3 independent experiments. **b** $^{13}$C labeled glucose was added to organoids derived from ACC1$^{\Delta/\Delta IEC}$ and ACC1$^{lox/lox}$ mice directly after 4-OHT treatment. Organoids were harvested at indicated time points and incorporation (δ) of $^{13}$C into de novo synthesized fatty acids was measured by mass-spec. Data pooled from N = 3 independent experiments. **c** Accumulation of neutral lipids in organoids. LipidTox green was added 30 min prior to analysis of mean fluorescence intensity (MFI) by flow cytometry. Data is representative for 4 independent experiments. **d** Western blot analysis of nuclear PPARδ and β-catenin protein was performed at indicated time points in organoids derived from ACC1$^{lox/lox}$ and ACC1$^{\Delta/\Delta IEC}$ mice. Anti-histone H3 was used as loading control. Representative data is shown from one out of 2 independent experiments with similar results. **e** ACC1$^{\Delta/\Delta IEC}$ and ACC1$^{lox/lox}$ control mice were injected with tamoxifen and treated daily (i.p.) with PPARδ agonist GW501516 or vehicle (veh) until analysis on day 14. H&E stainings of ileum sections are representative of 2 independent experiments with n = 3–5 mice per group. Scale bar represents 100 µm. **f–i** Organoids from ACC1$^{\Delta/\Delta IEC}$ mice were cultured in the absence or presence of 4-OHT for 24 h to induce ACC1 deletion. PPARδ agonist GW501516 or vehicle control (DMSO) were added for the whole culture period of 5 days. **f** Representative pictures of organoids and quantification of crypt domains. More than 30 organoids per group were analyzed. GW = GW501516. Bar = 200 µm. **g** qPCR analysis of Lgr5 expression. **h** Western blot analysis of nuclear PPARδ and β-catenin protein. GW = GW501516. Representative data is shown from one out of 2 independent experiments with similar results. **i** Number of secondary organoids per dissociated crypt-derived primary organoids. Primary organoids from ACC1$^{\Delta/\Delta IEC}$ mice were cultured +/− 4-OHT for 24 h. GW501516 or DMSO were added directly after plating. Primary organoids were subcloned on day 4. Data was pooled (N = 3) **g, i** or is representative (**f**) for 3 independent experiments with similar results. Statistical significance was analyzed using Bonferroni-Dunn method (**b**), unpaired two-tailed Student's t-test (**c**, Benjamini–Hochberg procedure (**a**) or One-way ANOVA with Tukey's multiple comparison test (**f, g, i**) with *$p < 0.05$, **$p < 0.01$, ***$p < 0.001$. ****$p < 0.0001$. Exact p values provided as Source Data. Bar graphs represent mean and error bars indicate SD. Boxes in boxplots denote 25th to 75th percentiles with whiskers representing min-max, and the central line the median. Source data are provided as a Source Data file.

de novo FAS, can rescue the loss of ACC1. Importantly, activation of PPARδ did not lead to a complete rescue of the ACC1-deficient phenotype in organoids, which indicates that the mechanisms by which de novo FAS supports ISC function might be more complex. Palmitoylation is a very common post-translational modification of proteins. Up to date, several hundreds of proteins have been identified that are attached covalently with palmitate[35]. We show here that inhibition of de novo FAS in paneth cells does not interfere with their ability to support ISC function. This finding suggests that functional palmitoylation of Wnt ligands in paneth cells is thus not affected by loss of de novo FAS. Nonetheless, modulation of de novo FAS may affect the palmitoylation of other proteins. In that context, inhibition of FAS has been shown to affect palmitoylation of mucin proteins[36]. Moreover, a recent study showed that auto-palmitoylation of TEAD-proteins contributes to function of ISCs[37]. Notably, de novo FAS also contributes to the regulation of cellular acetyl-CoA levels, which can affect protein acetylation and may govern the fate of stem cells by modulating epigenetic histone acetylation[16,38].

Nutrition can have a profound effect on stem cell biology. It has been demonstrated that ISCs respond to caloric restriction[7,10] and diet-derived factors including glucose[39], ketone bodies[8], and fat[11]. Although ISCs are able to take up fatty acids from the environment, our data suggests that under physiological conditions ISCs need to produce their own fatty acids to fulfil their needs. This dependence was already obvious in mice that were fed with a standard chow covering ~10% of energy intake by fat, which corresponds to the upper limit of fat intake recommended for humans by the U.S. Department of Agriculture[40]. The dependence on de novo FAS however became obsolete after increasing the fat content of the diet to >40% of energy intake, which is usually considered as high-fat or western-style diet. Together, these findings highlight the fact that the metabolic programming of ISCs, which under normal dietary conditions relies on the de novo production of fatty acids, is efficiently tuned by the composition of the diet and an excess delivery of environmental fat. Intriguingly, we observed that the distal parts of the both colon and small intestine were stronger affected by the loss of ACC1-mediated de novo FAS, while in particular the proximal colon showed only a mild phenotype. This might reflect the presence of sufficient external sources of fatty acids in the standard chow, which may be available in higher concentrations in

the proximal part of the small intestine. It is also possible that the microbiota in the proximal part of the colon can serve as a source of fatty acids. The presence of external sources of fatty acids may in fact also explain some of the differences that we observed in our in vitro and in vivo analyses, such as a significant increase in the expression of genes associated with secretory and absorptive enterocytes in vitro, but not in vivo. Nevertheless, we can also not rule out that in vivo some ISCs escape ACC1 deletion, in particular when the deletion affects the function and survival of ISCs, as it has been for example reported recently in mice with a conditional deletion of Hsp60 using the villin-creERT2 system[41].

The augmentation of ISC numbers and functionality by a pro-obesity HFD has been linked to the enhanced susceptibility towards colon cancer[11]. Our results show that reducing the activity of ACC1-mediated de novo FAS can restrict colonic tumor initiation in vivo, which is in line with previous results of ACC1 inhibition in a colon cancer cell line and a report suggesting a specific effect of FAS inhibition on cancer stem cells[42,43]. Whether this can be mechanistically linked to the effects on the PPARδ pathway, as it has been suggested in the case of HFD treatment[28], or to other effects of de novo FAS restriction, remains to be elucidated in future studies. Nevertheless, the results of our study point towards a significant role of ACC1-meditated de novo FAS in colon cancer development and highlight a potential impact of ACC1 inhibition in colon cancer prevention and therapy.

## Methods

All research conducted for this study complies with the relevant ethical regulations. Study protocols were approved by the Lower Saxony Committee on the Ethics of Animal Experiments as well as the responsible state office (Lower Saxony State Office of Consumer Protection and Food Safety) for all animal experiments and by the ethics committee of the Hannover Medical School for all work with human patient material.

**Mouse models.** Tamoxifen-inducible intestinal epithelium-specific ACC1-deficient mice (ACC1$^{\Delta/\Delta IEC}$) were generated by crossing Villin-Cre$^{ERT2}$ mice[18] with mice harboring loxP-flanked ACC1 alleles[17]. Lgr5-EGFP-IRES-Cre$^{ERT2}$ mice have been described before[4]. Tamoxifen-inducible Lgr5-specific ACC1 deficient mice (ACC1$^{\Delta/\Delta Lgr5}$) were obtained by crossing Lgr5-EGFP-IRES-Cre$^{ERT2}$ with ACC1$^{lox/lox}$ mice. All mice were bred on C57BL/6 J background and kept under specific pathogen free condition in individually ventilated cages (IVC) at the animal facilities of TWINCORE (Hannover, Germany) or the Helmholtz Centre for Infection Research (Braunschweig, Germany). Animals were handled in accordance with good animal practice as defined by FELASA and the national animal welfare body GV-SOLAS guidelines. All animal experiments were performed under the approval by the Lower Saxony Committee on the Ethics of Animal Experiments as well as the responsible state office (Lower Saxony State Office of

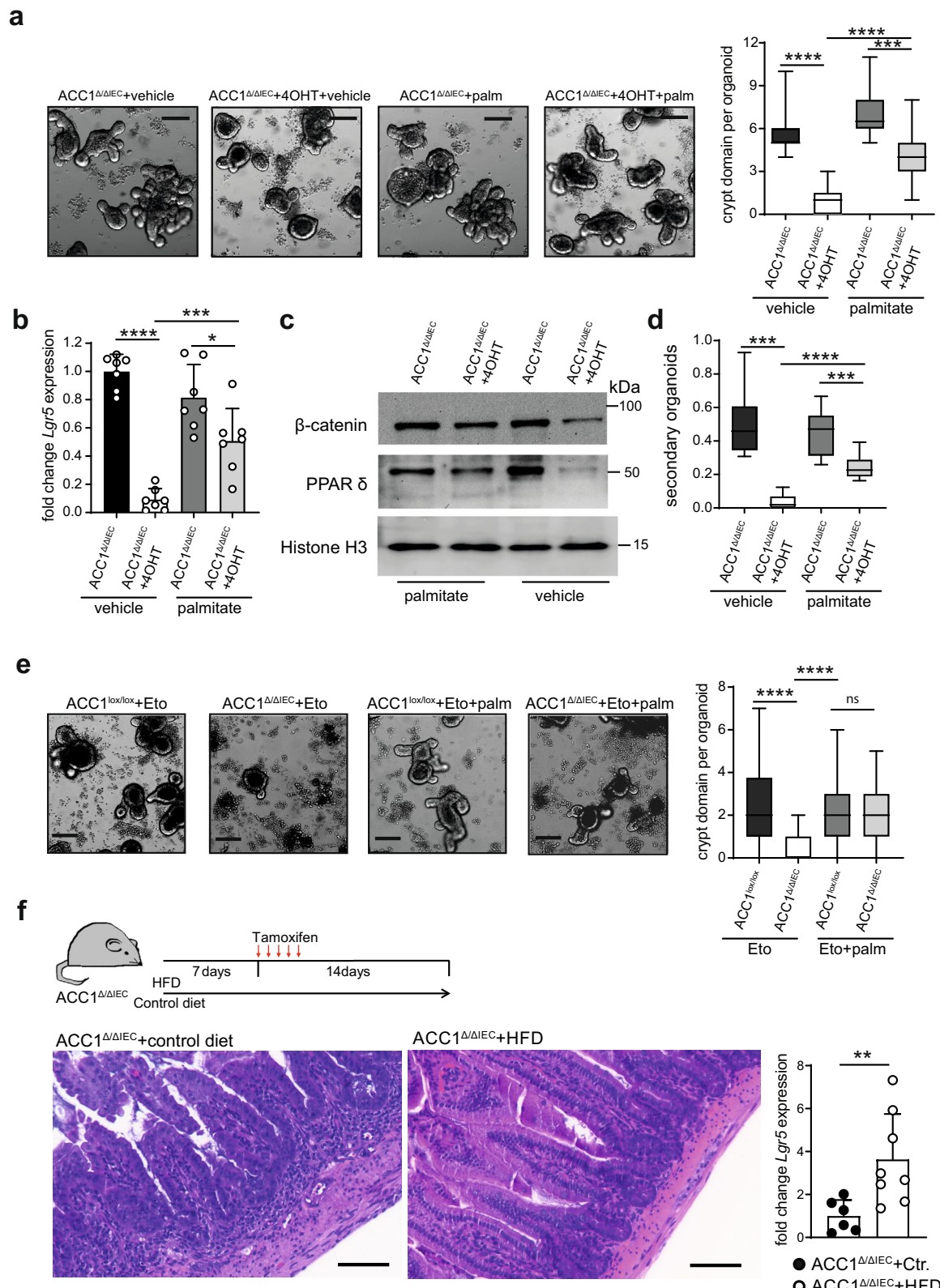

Consumer Protection and Food Safety) under the permit number 33.9-42502-04-16/2329 and 33.9-42502-04-15/1851. For all in vivo experiments, sex-matched 8–14 weeks old animals were used.

In vivo deletion of ACC1 was induced by intra peritoneal (i.p.) injection of 100 µl tamoxifen (Sigma, 10 mg/ml) suspended in corn oil (Sigma) for 5 consecutive days in ACC1$^{\Delta/\Delta IEC}$ mice. ACC1$^{\Delta/\Delta Lgr5}$ mice were treated every

second day with tamoxifen (i.p.) for 20 days as described previously[8] and analysed one day after the last injection.

For in vivo activation of PPARδ, the PPARδ-agonist GW501516 (Enzo Life Sciences) was reconstituted in DMSO at 20 mg/ml and diluted in a solution of 5% PEG400 (Sigma), 5% Tween80 (Sigma) and 90% water for daily intraperitoneal injections at 5 mg/kg body weight.

**Fig. 5 External lipid delivery compensates for the lack of ACC1-mediated de novo FAS. a–d** Crypts were isolated from ACC1$^{\Delta/\Delta IEC}$ mice and grown in organoid cultures in the absence or presence of 4-OHT for 24 h to induce ACC1 deletion. Palmitate was added into cultures with 100 µM on day 3 and analyzed 48 h later. **a** Organoids were imaged and the number of crypt domains was quantified. Crypt domain formation was calculated from >30 organoids per group. (Bar = 200 µm). **b** qPCR analysis of *Lgr5* expression. **c** Western blot analysis of nuclear PPARδ and β-catenin protein. **d** Secondary culture of organoids from palmitate-treated primary culture. **e** Organoid cultures of crypts isolated from ACC1$^{\Delta/\Delta IEC}$ and ACC1$^{lox/lox}$ mice were treated with Etomoxir (Eto, 10 µM) and palmitate (palm, 100 µM) or vehicle directly after 4-OHT treatment. Crypt domain formation was calculated at day 4 from >30 organoids per group. Data was pooled (N = 3) (**a**, **b**, **d**, **e**) or is representative (**c**) for 3 independent experiments with similar results. **f** ACC1$^{\Delta/\Delta IEC}$ mice were injected with tamoxifen and fed with a high fat (HFD) or control diet as indicated in the schematic overview. H&E stainings of ileum sections and expression of *Lgr5* gene in IECs isolated from the small intestine. Data shown are representative of 3 independent experiments with n = 3–5 mice per group. Scale bar represents 100 µm. Statistical significance was analyzed using unpaired two-tailed Student's t-test (**f**) or One-way ANOVA with Tukey's multiple comparison test (**a**, **b**, **d**, **e**) with *$p < 0.05$, **$p < 0.01$, ***$p < 0.001$. ****$p < 0.0001$. Exact p values provided as Source Data. Bar graphs represent mean and error bars indicate SD. Boxes in boxplots denote 25th to 75th percentiles with whiskers representing min-max, and the central line the median. Source data are provided as a Source Data file.

Short-term HFD was performed by feeding mice a diet containing 42% kJ of total energy derived from fat (Western style diet food TD.88137, SNIFF), starting one week before tamoxifen treatment for a total period of three weeks. Control mice were fed with the respective control diet recommended by the vendor (CD.88137, SNIFF, 13% kJ from fat).

For the Azoxymethane (AOM) / dextran-sodium-sulfate (DSS) model of colorectal tumorigenesis, ACC1$^{\Delta/\Delta IEC}$ or control ACC1$^{lox/lox}$ mice were injected i.p with tamoxifen for 5 consecutive days and treated on day 6 with AOM (Sigma, 10 mg/kg dissolved in sterile PBS, i.p.). After 5 days, mice were treated with 2% DSS (MP Biomedicals) in their drinking water (1st DSS cycle) for 5 days, followed by a rest of 2 weeks receiving regular drinking water. The DSS treatment was repeated (2nd DSS cycle) followed by another 2-week rest. A second AOM dose (10 mg/kg, i.p.) was administered 1 day before the 3rd DSS cycle. Mice were sacrificed on day 16 after the 3$^{rd}$ DSS cycle.

**Intestinal crypt isolation and culture of primary murine organoids**. Mouse crypts were isolated from total small intestine and cultured in 3D Matrigel following the manufacturer's protocol (Mouse IntestiCult™ Organoid Growth Medium, Stemcell Technologies). In brief, mice were sacrificed and the small intestine was harvested. After removing feces and remaining fat, the small intestine was flushed 3–5 times with ice cold PBS using a syringe with a p200 pipette tip attached. After flushing, a longitudinal incision along the length of the intestine was performed. The intestine was held by one end over a 50 ml falcon tube containing 15 ml of cold PBS with 1% Penicillin/Streptomycin. The intestine was cut into 2 mm pieces, all pieces were collected and transferred into a 50 ml falcon tube. A pre-wetted 10 ml serological pipette was used to wash the intestinal pieces by pipetting up and down 3 times. Tissue pieces were washed 10 times with 15 ml of cold PBS. Afterwards, the tissue pieces were resuspended with 25 ml of Gentle Cell Dissociation Reagent (Stemcell Technologies) and incubated on a rocking platform at 20 rpm for 15 min at room temperature. After incubation, the supernatant was carefully removed. The tissue pieces were resuspended in 15 ml of cold PBS + 0.1% BSA, and pipetted up and down 3 times. When the majority of the intestinal pieces settled to the bottom, the supernatant (the first fraction) was transferred and passed through a 70 µm filter into a new 50 ml conical tube. Three additional fractions were obtained. Fractions were centrifuged at $300 \times g$ for 5 min at 4 °C and supernatant was discarded. The pellets containing the isolated crypts were resuspended with 20 mL of cold PBS + 0.1% BSA and were centrifuged at $200 \times g$ for 3 min. Crypts were subsequently seeded in Matrigel (Corning) and cultured in IntestiCult™ Organoid Growth Medium (Stemcell Technologies).

To delete ACC1 in organoids derived from ACC1$^{\Delta/\Delta IEC}$ mice, 300 nM 4-hydroxytamoxifen (4-OHT, Sigma) was added to the culture for 24 h. In some experiments the PPARδ-agonist GW501516 (Enzo Life Sciences, dissolved DMSO) was added into organoid cultures with a final concentration of 1 µM directly after plating. Palmitate (Sigma, complexed with BSA) was added in a final concentration of 100 µM to organoid cultures. Etomoxir (Sigma, E1905), dissolved in DMSO, was added into organoid cultures with a final concentration of 10 µM.

**Ex vivo clonogenicity assay and secondary organoid assay**. For ex vivo clonogenicity assay, crypts were isolated from the small intestine of tamoxifen treated mice as described above. Crypts were counted and seeded in matrigel as described. The formation of organoids was assessed upon 2–3 days of culture in standard organoid growth medium (Stemcell Technologies). For the secondary organoid culture, primary organoids were mechanically dissociated, washed twice with PBS and embedded in matrigel. Growth of secondary organoids in standard organoid growth medium (Stemcell Technologies) was assessed after 24 h of culture.

**Human organoid cultures**. Crypts were isolated from surgically removed human intestinal tissue using a modification of the procedure described in[44]. Briefly, the mucosa of colonic tissue was lifted off the muscle layer and cut into small pieces. After washing with PBS, tissue pieces were incubated with chelating buffer (10 mM

EDTA in PBS, pH8.0) for 90 min on ice with shaking to remove the epithelial layer. Isolated crypts were washed twice with PBS and seeded in Matrigel (Corning) with human IntestiCult™ Organoid Growth Medium (Stemcell Technologies) into a well of pre-warmed 24-well culture dish. Soraphen A (dissolved in DMSO) was added with a concentration of 1 µM. Controls were treated with DMSO vehicle. The study protocol conformed to the ethical guidelines of the 1975 Declaration of Helsinki and all clinical procedures were approved by the ethical committee of the Hannover Medical School (permit number 3082–2016). Each patient has given well-informed written consent. Patents did not receive compensation for their participation. Samples were collected from adult male and female patients receiving surgery for colon cancer or inflammatory bowel disease. All samples were prepared from healthy parts of the resection.

**Organoid culture from sorted Lgr5$^+$ ISCs and paneth cells**. Crypts were isolated from the small intestine of Lgr5-EGFP-IRES-Cre$^{ERT2}$ mice as described above. To obtain single epithelial cells, the crypt suspension was incubated with TrypLE Express (Invitrogen), including Y-27632 (10 µM, Abmole) and N-acetyl-L-cysteine (1 µM, Sigma) for 20 min at 37 °C. An equal volume of ice-cold DMEM/F-12 was added to the suspension, resuspended and passed through 20 µm filters. Cells were pelleted by centrifugation at $300 \times g$ for 10 min. Cells were labeled with CD24-PE (eBioscience M1/69 1:400). To determine frequencies of Lgr5$^{high}$ ISCs and Lgr5$^{int}$ progenitors, cells were analyzed directly by flow cytometry. For organoid cultures, ISCs were sorted as Lgr5-EGFP$^{high}$ CD24$^-$DAPI$^-$. Paneth cells were sorted as Lgr5-EGFP$^-$-CD24$^{high}$ SSC$^{high}$ DAPI$^-$. The sorted ISCs were either embedded alone (500 cells per well) or together with paneth cells (500 cells per well) in matrigel and cultured in growth medium consisting of 50% conditioned WENR medium derived from L-WRN cells (ATCC® CRL3276™) supplemented with Y-27632 (10 µM, Abmole), N-acetylcysteine (1 µM, Sigma), N2 supplement (1× Thermo Fisher Scientific), B27 supplement (1×, Thermo Fisher Scientific), EGF (50 ng/ml, Sigma), murine noggin (100 ng/ml, Peprotech) and human R-spondin 1 (1 µg/ml, R&D). After 7 days, medium was changed to IntestiCult™ Organoid Growth Medium, which was defined as differentiation medium to induce crypt domain formation. Soraphen A (dissolved in DMSO) was added into organoid cultures at a final concentration of 200 nM, either at the beginning of the culture in growth medium directly after embedding sorted ISCs in matrigel, or directly after changing to differentiation medium. Controls were treated with DMSO vehicle.

**LipidTox staining**. Organoids were harvested and dissociated in to single cells by TrypLE Express (Invitrogen) including Y-27632 and N-acetyl-L-cysteine, as described above. Cells were first stained with the LIVE/DEAD Fixable Dead Cell Stain Kit (Life Technologies/Thermo Fisher Scientific). After washing, LipidTox green from HCS LipidTox Phospholipidosis and Steatosis Detection Kit (Thermo Fisher Scientific) was added to samples 30 min before analysis by flow cytometry.

**$^{13}$C$_6$ Incorporation assay**. [U-13C6] glucose (1 mM; Cambridge Isotope Laboratories) was added into organoid cultures directly after 24 h of 4-OHT treatment. Organoids were isolated from matrigel at different time points (0 h, 6 h, 24 h, 72 h) and snap-frozen. To assess the incorporation of glucose-derived carbon into cellular fatty acids, cells were saponified by (methanol: 3.75 M NaOH C1:1, v/v, 1 hr, 100 °C) and then prepared for analysis on a gas chromatography-combustion-isotope ratio mass spectrometer (GC/C/IRMS) as described earlier[45] with the following changes: extraction with n-hexane/t-butylmethyl ether (1:1) and second methylation step with addition of 450 µl MeOH and 50 µl trimethylchlorosilane to dried sample. Fatty acid methyl esters (FAMEs) were separated by GC-MM (5975 C GC coupled to 7890 A MSD, Agilent Technology, Santa Clara, CA, USA) and a BP X5 column (30 m × 0, 25 mm ID, 0,25 µm film thickness, 5% phenyl-polysilphenylene-siloxane, SGE, Munich, Germany) (50 °C for 1 min, 4 °C/min to 250 °C, 20 °C/min to 300 °C and hold for 1 min; column flow 1.2 mL/min, He) and identified by retention time and mass spectra after co-injection of a FAME standard. GC/C/IRMS measurements were performed in triplicate on a GC/C/IRMS

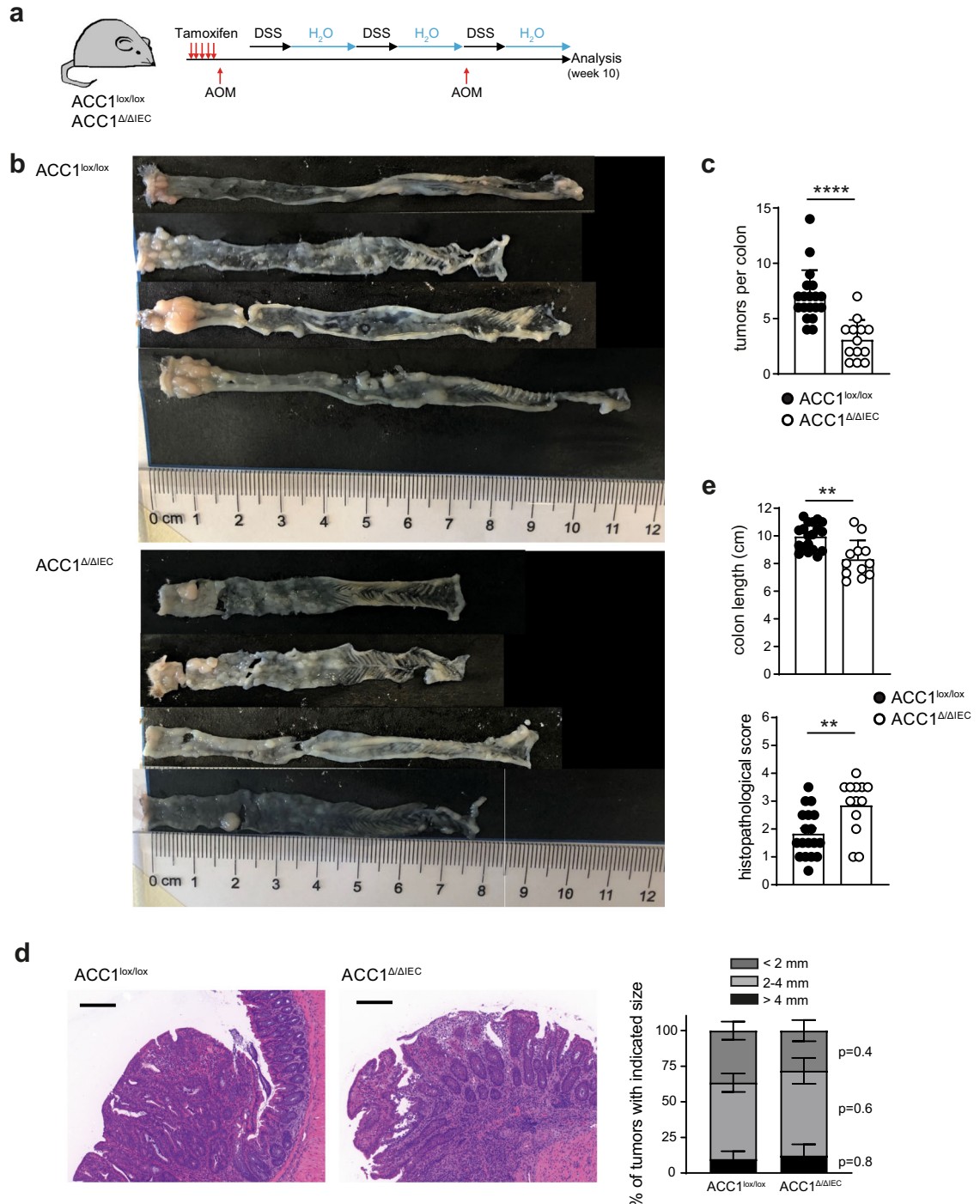

**Fig. 6 Epithelial inhibition of ACC1-mediated de novo FAS reduces inflammation-associated tumor formation. a** Experimental setting. Deletion of ACC1 was achieved through 5 consecutive days of tamoxifen injection (i.p). Mice were subsequently subjected to 3 cycles of DSS (2% w/v in drinking water) and 2 injections of AOM (10 mg/kg, i.p) before the 1st and 3rd DSS cycle. Analysis was performed on day 16 after the last DSS cycle, as depicted. **b** Pictures of the colons of ACC1$^{lox/lox}$ and ACC1$^{\Delta/\Delta IEC}$ mice are shown for one representative out of 3 independent experiments. **c** Numbers of macroscopically counted colonic tumors. **d** Representative H&E images of colonic tumors. Bar graph shows the distribution of colonic tumors identified in individual mice classified according to their size. Bar = 200 μm. **e** Measurement of colon length and the histopathological score. Data was pooled from 3 experiments with $n = 19$ (ACC1$^{\Delta/\Delta IEC}$) and $n = 13$ (ACC1$^{lox/lox}$) mice (**c, d**) and $n = 18$ (ACC1$^{\Delta/\Delta IEC}$) and $n = 13$ (ACC1$^{lox/lox}$) mice (**e**). Statistical analysis was performed using unpaired two-tailed Student's t-test with *$p < 0.05$, **$p < 0.01$, ***$p < 0.001$. ****$p < 0.0001$. Exact p values provided as Source Data. Bar graphs represent mean and error bars indicate SD (**c, e**) or SEM (**d**). Source data are provided as a Source Data file.

(7890 A GC system, Agilent Technology, coupled via ConFlo IV interface to a Finnigan MAT 253 Plus Isotope Ratio Mass Spectrometer, Thermo Fisher Scientific, Germany). The fatty acid methyl esters were separated with a BP X5 column (50 m, 0.32 μm ID, 0.25 μm film thickness). The oven program was 70 °C for 1 min, 20 °C/min to 130 °C, 2 °C/min to 150 °C/min and hold for 5 min, 2 °C/min to 165 °C and hold for 5 min, 20 °C/min to 300 °C and hold for 5 min. The separated compounds were combusted on line in an oxidation oven (copper, nickel catalyst, 1000 °C), reduced with elemental copper (600 °C) and finally dried by a water-permeable membrane (Nafion). 13 C/12 C isotope ratios for the free fatty acids were calculated as described[46] and are presented as δ13C in the figures.

**Isolation of intestinal lamina propria cells and flow cytometry**. Cells of the small intestinal and colonic lamina propria were isolated as described before[47]. Colon and small intestine were isolated from the animals, physically emptied, opened longitudinally, and cut into 2–3 cm pieces. Tissue pieces were incubated in PBS containing 30 mM EDTA (Roth) and washed vigorously to remove remaining mucus. Remaining tissue was further cut and digested in prewarmed Iscove's modified Dulbecco's medium (Life Tech- nologies/Gibco) containing 1 mg/mL Collagenase D (Roche) and 100 μg/mL DNaseI (Roche). The supernatant was fil- tered and the remaining tissue was passed through a 100 μm mesh. The cells were separated using a 40%/80% gradient (Percoll solution, GE Healthcare; 900 g, 20 min, 20 °C, no break). The interphase was harvested, washed, and cells were subsequently used for flow cytometry analysis according to the guidelines for the use of flow cytometry and cell sorting in immunological studies[48]. In brief, cells were first labeled with the LIVE/DEAD Fixable Dead Cell Stain Kit (Life Tech- nologies/Thermo Fisher Scientific) according to the manufacturer's recommen- dations. Cells were further incubated in PBS containing 0.2% BSA and 1% anti- mouse CD16/CD32 antibody (BioXcell) for 5 min on ice. Afterward, surface antigens were labeled in PBS containing 0.25% BSA (Roche), 0.02% NaN3 (Roth), and 2 mM EDTA (Roth) on ice for 30 min. Fluorescence-conjugated monoclonal antibodies that were specific to mouse antigens: CD11c-APC-eFluor780 (N418), Gr-1-eF660 (RB6-8C5), CD3-PerCP-Cy5.5 (145-2C11), CD19-eF450 (1D3), CD4-PE-Cy7 (GK1.5), and MHC Class II (I-A/I-E)-FITC (M5/114.15) from eBioscience/ Thermo Fisher Scientific, CD64(FcγRI)-PE (X54-5/7.1) from Biolegend, and CD45-PE-TexasRed (30-F11) from Invitrogen/Thermo Fisher Scientific.

**Gene expression analysis and RNA-Sequencing**. RNA from cells and organoids was prepared using RNeasy Micro Kit (Qiagen) according to the manufacturer's instructions. For RNA isolation from organoids, the organoid medium was replaced with ice–cold PBS and pipetted up- and down 20 times. RLT buffer (Qiagen) was added directly to the organoids pellets. After RNA isolation, 1 μg of total RNA was transcribed into cDNA using SuperScript III Reverse Transcriptase Kit (Thermo Fisher Scientific). All procedures were performed according to the manufacturer's protocols. Real-time PCR was performed using iQ SYBR Green Supermix (Bio-Rad). Primer sequences for the identification of IEC types and for assessing the deletion efficiency of ACC1 are specified in Supplementary Table 1. Gene expression was normalized to Actb and log2 transformed, and is shown as fold change.

For RNA-seq analysis, organoids were harvested as described above at 96 h upon 4-OHT treatment and total RNA was isolated using RNeasy Micro Kit (Qiagen) according to the manufacturer's instructions. Quality and integrity of total RNA was controlled on Agilent Technologies 2100 Bioanalyzer (Agilent Technologies). The RNA sequencing library was generated from 100 ng total RNA using NEBNext Single Cell/Low Input RNA Library Prep (New England BioLabs) according to the manufacturer´s protocols. The libraries were sequenced on Illumina NovaSeq 6000 using NovaSeq 6000 S2 Reagent Kit (100 cycles, paired end run) with an average of $3 \times 10^7$ reads per RNA sample. Each FASTQ file gets a quality report generated by FASTQC tool. Before alignment to reference genome each sequence in the raw FASTQ files were trimmed on base call quality and sequencing adapter contamination using Trim Galore! wrapper tool. Reads shorter than 20 bp were removed from FASTQ file. Trimmed reads were aligned to the reference genome using open source short read aligner STAR (https://code.google.com/p/rna-star/) with settings according to log file. Feature counts were determined using R package "Rsubread". Only genes showing counts greater 5 at least two times across all samples were considered for further analysis (data cleansing). Gene annotation was done by R package "bioMaRt". Before starting the statistical analysis steps, expression data was log2 transformed and TMM normalized (edgeR). Differential gene expression was calculated by R package "edgeR". Functional analysis (GSEA) was performed by R package "clusterProfiler".

**Histology and western blot**. For histological analyses, colonic and small intestinal tissues were sampled and fixed in Roti-Histofix (Roth) at the day of analysis. Samples were processed to paraffin blocks, sectioned, and stained with hematoxylin and eosin at the Pathology Department of the Hannover Medical School. For immunohistochemical analysis, antigen was first retrieved using Target retrieval solution (DAKO) according to the manufacturer's recommendations. Following antibodies were used to identify ISCs and paneth cells, respectively: Rabbit anti-OLFM4 (1:500, CST, 39141 T), and rabbit anti-MMP7 (1:100, CST, 3801 T). Biotin conjugated anti-rabbit IgG (1:700, Vector, PK6101) was used as secondary anti- body. The Vectastain Elite ABC immunoperoxidase detection kit (Vector Labs) was used for visualization. Histopathological scoring was performed using a scoring system described previously[49]. In brief, the presence of rare inflammatory cells in the lamina propria were counted as: 0, increased num- bers of inflammatory cells; 1, confluence of inflammatory cells; 2, extending into the submucosa; and 3, transmural extension of the inflammatory cell infil- trate. For epithelial damage, absence of mucosal damage was counted as 0, discrete focal lymphoepithelial lesions were counted as 1, mucosal erosion/ ulceration was counted as 2, and a score of 3 was given for extensive mucosal damage and extension through deeper structures of the bowel wall. The two subscores were added and the combined histological score ranged from 0 (no changes) to 6 (extensive cell infiltration and tissue damage).

For nuclear protein isolation, organoids were harvested from Matrigel by washing twice with ice-cold PBS. Ice-cold CER I solution was added to the pellets and further processed according to the manufacturer's instructions (NE-PER™ Nuclear and Cytoplasmic Extraction Kit, Thermo Fisher Scientific). Western blot was performed using Rabbit anti-β-catenin (1:1000, CST, 8814 S) and rabbit anti-PPARδ (1:500 Thermo Fisher Scientific, PA1-823A). Mouse anti-histone H3 (1:1000, CST, 4499 S) was used as loading control. Pierce™ ECL Western Blotting Substrate (Thermo Fisher Scientific) was used for development. Blots were imaged using a Chemo Star Professional Imager (Intas).

**Statistical analysis**. RT-PCR data was acquired using a LightCycler 480 II (Roche). Flow cytometry data was acquired on a LSR II (BD) and analyzed with FlowJo software (Tree Star). Cells were sorted on a FACSAria IIu, (Becton Dick- inson) at the Cell Sorting Core Facility of the Hannover Medical School. Histo- logical samples and organoids were imaged and processed with Nuance 2.10.0 software (Cambridge Research & Instrumentation). All histological analyses were performed by a blinded investigator. Statistics were analyzed using GraphPad Prism Software 8.4. No statistical method was used to predetermine sample size. Statistical significance of results was determined with two-tailed Student's $t$-tests, ANOVA or Benjamini–Hochberg procedure as indicated in the figure legends and no data were excluded from the analyses. Results shown as bar graphs represent mean ± SD. Boxes in boxplots denote 25th to 75th percentiles with whiskers representing min-max, and the central line the median. $P$-values considered sig- nificant as follows: $*p < 0.05$, $**p < 0.01$, $***p < 0.001$. $****p < 0.0001$, ns = not significant.

**Reporting summary**. Further information on research design is available in the Nature Research Reporting Summary linked to this article.

## Data availability
Sequencing data reported in this paper was uploaded to GEO: Accession number for the RNA-Seq data is GSE188386. Uncropped versions of the western blots shown in Figs. 4d, h and 5c have been provided as Supplementary material. Other data from the findings of this study are available from the corresponding author upon request. Source data are provided with this paper.

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

## Acknowledgements

This work was supported by grants from the German Research Foundation (DFG LO1415/7-1 to M.L.) and the state of lower Saxony (R2N). We thank Rolf Müller for providing Soraphen A. We would like to acknowledge the assistance of the Genome Analytics Facility of the Helmholtz Centre for Infection research and the Cell Sorting Core Facility of the Hannover Medical School supported in part by Braukmann-Wittenberg-Herz-Stiftung and Deutsche Forschungsgemeinschaft.

## Author contributions

S.L., C.L., E.D., W.L., M.G., M.L., F.K. and M.B. performed experiments. S.F., M.R.W., R.G., H.H.R., W.R.A. and G.A.G. provided essential materials and helped with interpretation of the data. S.L. and M.L. designed the study, interpreted the data and wrote the manuscript.

## Funding

## Competing interests

The authors declare no competing interests.
