## [Peer Review File · Nature Communications]

Acetyl-CoA-Carboxylase 1-mediated de novo fatty acid synthesis sustains Lgr5+ intestinal stem cell functionREVIEWER COMMENTS

Reviewer #1 (Remarks to the Author):

Li et al. demonstrated that disruption of de novo FAS by depleting ACC1 resulted in the loss of intestinal stem cells and crypt structures by utilizing Villin-CreERT2 mouse model. The phenotype (loss of crypts and ISC marker expression) is very interesting, indicating a critical role for ACC1. Unfortunately, there are some inconsistencies or areas where rigor could be improved with this work (replicates in RNAseq, use of potentially toxic inhibitor, use of organoids where in vivo would be more appropriate, lack of metabolomics confirmation of disruption of FAS in the model). Additionally, the authors do not propose a mechanism through which ACC1 could be required for ISC renewal.

The authors suggest that for better understanding of ACC1 function in the intestinal stem cells, ISC specific Cre, Lgr5-EGFP-IRES-CreERT2 should be used in this study, especially when it is already available in the lab and used in the sorting experiments. The mosaic Lgr5-CreERT2 driver could provide a good model to study the function of ACC1 in the stem cells, with its internal controls. It is interesting that palmitate and HFD rescued ACC1 knockout phenotype in vitro and in vivo, respectively, indicating the extra lipid enriched supply can sustain ISC function when de novo FAS is compromised. However, the authors investigated the involvement of PPAR δ / β -catenin pathway but didn't go further for the mechanisms. How does ACC1 mediated de novo FAS interact with PPAR δ / β -catenin pathway?

Taken together, I respectfully feel that the proposed work falls short of what's expected for Nat. Comms.

Major:

Do authors have data for ACC1 KO by using the Lgr5-EGFP-IRES-CreERT2 instead of Villin-CreERT2?

Is there any ACC1 IHC/IF staining in the WT and KO? Is ACC1 specifically enriched in ISCs, or general expressed in the whole intestinal epithelium?

Is FAS disrupted in these mutants? Can the authors demonstrate this using metabolomics? Could another enzyme help compensate for loss of ACC1? Or might the knockout be incomplete?

Similarly, for B-catenin, a western is shown, indicating a dramatic reduction in protein levels upon ACC KO, yet is this difference seen in IHC? It's surprising that you would have such a decrease in B-catenin, given that most of the protein localized at the membrane and doesn't participate in signaling.

If the ACC1 KO mice lost their ISCs and crypt structures (2 weeks after TAM), where do villi cells come from? It is surprising that these ACC1 KO mice didn't seem to show weight loss or overt signs of disease with such an obvious histological change in the intestine. Why?

When does the crypt loss phenotype start in ACC1 KO? Do authors have data at early time point after TAM treatment and long-term observations for these ACC1 KO? Does crypt loss also occur in duodenum and jejunum, as authors only show the phenotype in ileum and colon in the figures? It is rather surprising that the animals can survive with this much crypt damage? Is the knockout partially efficient and the epithelium recovers over time with ACC1-WT cells? What cells remain proliferative to maintain villus epithelial turnover in the mutant?

The authors documented that quiescent Tert⁺ ISC subpopulation is less affected in ACC1 KO in vivo first, but then showed a significant alteration of Tert in vitro (Fig 3d). Explanation or discussion for these contradictory results in vivo vs in vitro is need.

The authors documented that they used 10 μ M Soraphen A in the method section, but claimed they added 200 nM Soraphen A in figure legend of Fig. 3f. Could the authors clarify about that?

What is the concentration of Soraphen A used in Fig 3e? Is it the same as the one used in Fig 3f?

How do authors know the inhibition of organoid formation is due to inhibition of de novo FAS (Soraphen acts as an inhibitor) instead of the toxicity of Soraphen A itself? If the mice are treated with Soraphen A in vivo, will the mice show similar phenotype as ACC1 KO?

The authors analyzed gene expression in organoids instead of in vivo tissues, and claimed that few differences were found between ACC1 KO and controls. The organoid in vitro system is bringing more artificial growing conditions and organoids are growing in heterogeneity, when the in vivo tissues are available, gene expression analysis could be performed on in vivo tissues, such as the

isolated ISCs, or crypts. The differences in organoid morphology in Fig. 3 suggest more differences should be observed in the RNAseq in Figure 4. Also, more than 2 replicates should be performed in RNAseq.

In Fig 4g, besides ACC1 KO, are we also expecting to see increased PPAR δ expression with GW501516 in controls (seems to be down in controls, GW vs DMSO)? It seems PPAR δ agonist also compromises stem cells in controls (Lgr5 expression in Fig 4f), why?

Can WNT agonists also rescue the ACC1 KO organoids?

FAS in ACC1 KO vs controls should be an important evaluation in this study. The authors should design more experiments to investigate the function of ACC1 in FAS in the intestine. For example, authors could compare lipid profiles of ACC1 KO vs controls (in vivo samples, early time point after TAM, before crypt structure loss) and investigate fatty acid synthesis in ACC1 KO vs controls by incorporating labeled acetate in organoids.

The C16 saturated fatty acid palmitate is the end product of de novo FAS, and the authors demonstrated that palmitate can rescue the phenotype of ACC1 KO. How about the effects of unsaturated fatty acid or other fatty acids (short chain, medium-chain, etc) in these ACC1 KO? Do ACC1 KO completely lose the function of de novo FAS? Can extra acetate (a precursor of acetyl-CoA) also rescue the phenotype of ACC1 KO?

Minor:

The ACC1 KO panel in Fig 1b seems to be cross-section tissues. It is hard to see the clear morphology of intestine in the KOs.

In Fig 1c, any thoughts about why increased frequencies of innate leukocytes were only observed in the small intestine, but not in the colon?

The legend key is missed in Fig 2c, Fig 3d panel, although we can guess from other panels.

Is Fig 6b a representative figure? The tumor size is much bigger in ACC1 KO mice compared to controls. It does not fit for the distribution of tumor size panel, Fig 6e.

Reviewer #2 (Remarks to the Author):

The authors characterize the intestinal phenotype in mice that lack ACC1, a protein that is involved in de novo fatty acid synthesis. The authors have previously explored the function of de novo fatty acid synthesis and ACC1 in immune T cell populations, while here the authors characterize the role of ACC1 within intestinal stem and progenitor populations, proliferating populations that need lipids for cellular division. Overall, we find the story interesting yet preliminary in its analysis and characterization of the phenotype, and, importantly, the tumor phenotype requires more clarification.

Specific comments:

Given that ACC1 function is essential and that this is first characterization of ACC1 loss-of-function in the intestine it would be informative to characterize the kinetics of the phenotype. How early does the phenotype present? How long do adult intestinal KO animals live? Two weeks of deletion is an early timepoint and overt signs of disease may not have time to manifest. At longer timescales of excision, are disease pathologies present?

Do you only achieve 90% excision efficiency because of epithelial escapers or are non-epithelial cells contaminating your sample?

The epithelia appears severely damaged, please characterize further. Is there increased apoptosis or necrosis? Is the tissue still proliferating? Is there a change in the villi length, please quantify. Please quantify the number of crypts per a given length of intestine. The authors should better characterize the intestinal epithelial phenotype by quantitating the following: cell-death/apoptosis, proliferation, ISC fate mapping, in vivo ISC numbers, and survival (ie survival curve).

The authors should also provide clarification regarding the metabolic adaptation that occurs when de novo FAS is dampened (ie compensation by ACC2, FASN) For instance, proliferating cells use

lipids to divide so how do ISCs and progenitors adapt? Are the cells recycling lipids or are they cannibalizing lipid stores? Some evidence or discussion of these points would improve the manuscript.

It is surprising that changes in intestinal cell types does not change. The MMP7 marker for Paneth cells looks more broadly expressed in ACC1-KO crypts compared to WT controls. Please quantify the histology. Also, the ACC1KO tissue for the PAS staining looks inconsistent with the MMP7, the crypts look very large and deep, hypertrophic.

What is the frequency of Lgr5+ cells and Olfm4+ cells per crypt in different regions of the intestine? It is important to point out that Olfm4 marks not only ISCs but early progenitors and is thus more broadly expressed compared to Lgr5 in the crypt.

In figure 3D, the levels of Muc2, Lyz1, and Chgb all appear to be increasing in the knock-out. Why is this the case? Is a loss of ACC1 pushing cells into a secretory, post-mitotic state in order to conserve lipids needed for division?

In their organoid assays, the authors delete ACC1 in propagating organoids in culture. However, it is also important to test stem cell function using the organoid assay by deleting ACC1 in vivo, prior to culturing, and then to quantify organoid clonogenicity as measure of stem cell activity. Further characterization of the organoids is also desired; for example, are the ACC1-null organoids proliferating and growing in size or what is the % live cells of organoids? The method the authors use allows organoids to form in the presence of residual ACC1 functional mRNA and protein. Crypt domains in organoids are an indication of differentiation; markers of differentiation increase in ACC1-null crypts, however, the crypt budding domains decrease. How do the authors explain this discrepancy?

The authors use a "growing medium" that contains WNT and RSPO. Why does this media not overcome the deficiency of b-catenin levels (WNT activation) as noted in Figure 4? A concentration curve with increasing recombinant Wnt3a to test the role of Wnt signaling in rescuing the growth and budding defects would be helpful.

Why does PPARdelta activation rescue -what targets are activated? Does PPARdelta agonist rescue the in vivo phenotype? Is there an alternate route to making lipids other than through ACC1, such as compensation by ACC2 or FASN, or increased recycling of intracellular lipids? Can acetate rescue the ACC1-null organoids? In the organoid assays for GW/Palmitate/HFD, does less organoid death occur? Please quantify clonogenicity. Separately, PPARgamma is associated with lipogenesis, perhaps the authors could also check this PPAR family member. It would be informative to know how other PPAR family members (delta, alpha and gamma) are impacted by ACC1 loss. Can PPARgamma activation rescue the phenotype?

What are the genes in the heat map? Please annotate. Why does it appear that the ACC1-WT genes flip in expression from day1 to day4? Please clarify the headings in the supplemental table, Currently, it is unclear what the column headings indicate. Also, in general Ppar levels change very little upon stimulation. A better readout of PPARdelta activity is its transcriptional targets.

The tumor data is not very strong. The ACC1 has larger fractions of bigger tumors but fewer total tumors indicating that perhaps tumors are not clonal and have fused. These data require more analysis and perhaps assessing tumor burden at earlier time points would help.

Minor:

Consider normalizing the QPCR data to control.

Sup fig 3a schematic is not consistent with Fig6a.

Figure 3D needs significance calculated

Error bars are needed on Supp Fig 1

Clarification is needed in lines 136-138 as to which cell type is being measured.

Reviewer #3 (Remarks to the Author):

In this manuscript, the authors investigate the role of ACC1 in the fatty acid synthesis pathway in maintaining Lgr5+ intestinal stem cell function. While these findings are interesting and worth exploring, critical conclusions from this study are not sufficiently supported by the presented data. More detailed comments are as follows:

Major comments:

1. In the de novo fatty acid synthesis pathway, ACC1 mediates acetyl-CoA conversion to malonyl-CoA, then FAS mediates the malonyl-CoA to generate the end product Palmitate. Inhibition or deletion of ACC1 blocks the fatty acid synthesis pathway. Does extra acetyl-CoA drive beta-oxidation in all the models, therefore driving the intestinal stem cell proliferation?

In addition, malonyl-CoA also plays an important role in the regulation of fatty acid oxidation in the mitochondria as an inhibitor of carnitine palmitoyl transferase 1, which performs the first step in the transfer of long-chain fatty acyl CoA into mitochondria for their oxidation.

Although the authors generated intestinal organoids from ACC1-deleted IEC mice and observed only minor effects on OCR and no critical impact on mitochondrial metabolism, why are the basal OCR levels of ACC1 IEC KO mice higher than the ACC1 flox mice? Are there any controls (cell number or protein amount) for the Seahorse organoid assay? Is this experiment carried out on day 1 or day 4?

2. The authors found increased numbers of innate leukocytes such as DCs and Macrophages in the small intestine but not in the colon of ACC1 KO mice by flow cytometry in Figure 1b. ACC1 KO mice were also challenged with AOM-DSS and the authors concluded that inhibition of ACC1 reduced the formation of tumors. While the tumor load decreased in Figure 4c, both Figure 4b and 4e show that tumor sizes increased in ACC1 IEC KO mice. In addition, reduced colon lengths in Figure 4b and 4d both indicated more severe inflammation in ACC1 IEC KO mice. Does ACC1 contribute to tumor progression or not?

All other data presented are ACC1 deletion in healthy states, such as ACC1-mediated PPARdelta/b-catenin activation in ISCs.

Detailed comments:

1. Figure 1a, significance should be indicated.

2. Figure 1b, the small intestine section of ACC1 KO IEC mice is cut at an angle and therefore cannot be used to count villi length or crypt shapes. Scale bars are missing.

3. Figure 1c, the Y axis goes directly from 10 to 100. It is difficult to determine the T, B and Macrophage percentages from the graph. The gating strategy could be put into Supplementary data, and real flow images with X and Y axis comparing ACC1flox and ACC1KO mice should be presented in the main figure.

4. Figure 2, since authors have Lgr5-EGFP mice, have they examined GFP+ cells in the ACC1 KO mice? How about EdU incorporation in ACC1 KO mice?

5. Figure 2c, why is the Y axis so low, relative expression to which gene? Any explanation why Tert gene or quiescent subpopulation of ISCs does not change in ACC1 KO mice? Does it also not change in AOMDSS injury model?

6. Figure 3b, when you knockout ACC1, what happens to FAS, ACC2 expression?

7. Although in Figure 4 the authors claimed ACC1 contributions to ISC function is dependent on PPARdelta/b-catenin pathway, no PPARdelta pathway genes such as Cpt1a, Angptl4, Pdk1, Hmgcs2 etc., show significant expression changes (Supp table 1). Please explain this discrepancy.

8. Figure 3b, while the expression of Lgr5 and Tert are not increased in ACC1KO organoids from day 1 to day 5, the expression of Muc2, Lyz1 and Chgb appears to gradually increase in ACC1KO organoids from day 1 to day 5. Please explain this discrepancy.

9. Figure 3f: The scale bar is missing. Is the SorA-treated organoid placed in growing medium at day 1? Would be helpful to include both low and high magnification images to show the sizes of each one.

10. Figure 4d, again GW1516 activates PPARdelta to elicit the beta-oxidation pathway. Therefore, one cannot conclude from this data that ACC1 contribution to ISC maintenance is FAS-dependent.

11. A similar question with palmitate and HFD compensation. What happens to the OCR profile when palmitate is added to ACC1 KO organoids?

12. Figure 5. As mentioned above, ACC1 KO mice upon AOMDSS challenge seem to have severe inflammation and increased tumor burden rather than reduction of tumorigenesis.

We would like to address the referees' questions as follows:

Reviewer 1

Li et al. demonstrated that disruption of de novo FAS by depleting ACC1 resulted in the loss of intestinal stem cells and crypt structures by utilizing Villin-CreERT2 mouse model. The phenotype (loss of crypts and ISC marker expression) is very interesting, indicating a critical role for ACC1. Unfortunately, there are some inconsistencies or areas where rigor could be improved with this work (replicates in RNAseq, use of potentially toxic inhibitor, use of organoids where in vivo would be more appropriate, lack of metabolomics confirmation of disruption of FAS in the model). Additionally, the authors do not propose a mechanism through which ACC1 could be required for ISC renewal. The authors suggest that for better understanding of ACC1 function in the intestinal stem cells, ISC specific Cre, Lgr5-EGFP-IRES-CreERT2 should be used in this study, especially when it is already available in the lab and used in the sorting experiments. The mosaic Lgr5-CreERT2 driver could provide a good model to study the function of ACC1 in the stem cells, with its internal controls. It is interesting that palmitate and HFD rescued ACC1 knockout phenotype in vitro and in vivo, respectively, indicating the extra lipid enriched supply can sustain ISC function when de novo FAS is compromised. However, the authors investigated the involvement of PPAR δ / β -catenin pathway but didn't go further for the mechanisms. How does ACC1 mediated de novo FAS interact with PPAR δ / β -catenin pathway? Taken together, I respectfully feel that the proposed work falls short of what's expected for Nat. Comms. Major:

1. Do authors have data for ACC1 KO by using the Lgr5-EGFP-IRES-CreERT2 instead of Villin-CreERT2?

Author response: We thank the reviewer for this important suggestion. We have crossed ACC^{lox/lox} mice with Lgr5-EGFP-IRES-Cre^{ERT2} mice to allow for the specific deletion of ACC1 in Lgr5⁺ ISCs (ACC1 ^{Δ /Lgr5} mice). Ex vivo analysis of Lgr5⁺ cell populations revealed significantly reduced frequencies of Lgr5 high expressing ISCs in the lower part of the small intestine, as well as in the colon of ACC1 ^{Δ /Lgr5} mice. In crypts isolated from both locations we also observed a reduction in the frequencies of Lgr5 intermediate expressing cells, which most likely represent direct progenitors of ISCs. Thus, our data using ACC1 ^{Δ /Lgr5} mice confirm a cell-autonomous role for ACC1 in Lgr5⁺ ISC. The data is shown in the novel **Figure 2e** and **Supplementary Figure 2**).

2. Is there any ACC1 IHC/IF staining in the WT and KO? Is ACC1 specifically enriched in ISCs, or general expressed in the whole intestinal epithelium?

Author response: We agree with the reviewer that direct staining of ACC1 would be optimal. Yet, direct comparison of ACC1 protein in WT versus KO is difficult, since the ACC1-deficient cells still express a truncated (enzymatically inactive) protein that lacks exon 22 of the ACC1 protein (doi:10.1073/pnas.0603115103). To the best of our knowledge, no antibody is available that would allow for the specific staining of the inactivated protein. Although we have not systematically tested ACC1 expression levels in the different epithelial cell types, our qPCR analysis point towards expression in the whole epithelium. We would like to point out that regulation of de novo FAS seems to be rather not on the level of transcription, as indicated by our own new data (**new Figure 4a**), as well as by other studies which showed that interference with de novo FAS did not result in major changes in the transcription-levels of de

novo FAS-associated genes (Doi:10.1038/nature11689). This is also in accordance with the idea that ACC1 activity is mainly regulated on the post-transcriptional level, e.g. by phosphorylation-induced multimerization or allosteric activation or inhibition (Doi.org/10.1016/j.it.2014.12.005). Independent of these considerations, we believe that our new data using ACC1^{Δ/ΔLgr5} mice demonstrate that ACC1 is of particular importance for ISCs.

3. Is FAS disrupted in these mutants? Can the authors demonstrate this using metabolomics? Could another enzyme help compensate for loss of ACC1? Or might the knockout be incomplete?

Author response: We agree with the reviewer that it is important to confirm the direct impact of ACC1-deletion on de novo FAS. To do so, we have now performed ¹³C-based flux assays to follow directly the synthesis of fatty acid synthesis in WT and ACC1-deficient organoids. While we observed in WT a gradual increase of glucose-derived ¹³C accumulation into palmitate and the derivatives stearate and oleate, our results show clearly that de novo FAS is completely disrupted in ACC1-deficient organoids (**new Figure 4b**). Of note, we found that addition of acetate does not lead to a rescue of the ACC1-deficient phenotype (**new Supplementary Figure 4b**). This finding is in accordance with the idea that ACC1 is indeed essential for the de novo generation of fatty acids from the precursor acetyl-CoA. Our data also indicate that there is no significant functional compensation for the lack of ACC1, e.g. by the activity of ACC2 or upregulation of Fasn. In summary, we believe that our findings confirm the efficient disruption of denovo FAS in epithelial cells as a direct consequence of ACC1-deletion.

4. Similarly, for B-catenin, a western is shown, indicating a dramatic reduction in protein levels upon ACC KO, yet is this difference seen in IHC? It's surprising that you would have such a decrease in B-catenin, given that most of the protein localized at the membrane and doesn't participate in signaling.

Author response: We appreciate the concerns raised by the reviewer. We would like to point out that we have chosen to directly assess the nuclear amounts of β-catenin. In the absence of Wnt signaling (off state), β-catenin is arrested in the cytosol by the APC/Axin/GSK-3-complex, leading to proteasomal degradation. Upon Wnt ligand binding to its receptor (on state), β-catenin translocates into the nucleus and binds to the TCF transcription factors, which induces Wnt target gene expression. Therefore, we respectfully feel that the specific determination of nuclear β-catenin protein level by western blot is an appropriate (and in our view also widely accepted) correlate of the activity of the Wnt-signaling pathway.

5. If the ACC1 KO mice lost their ISCs and crypt structures (2 weeks after TAM), where do villi cells come from? It is surprising that these ACC1 KO mice didn't seem to show weight loss or overt signs of disease with such an obvious histological change in the intestine. Why?

Author response: In order to better define the consequences of IEC-specific ACC1-deletion, we have now included an extended analysis of the histological phenotype. Our analysis shows that the distal parts of the both colon and small intestine are stronger affected by the loss of ACC1, while in particular the proximal colon shows only a mild phenotype (**new Figure 1b** and **c**). This phenotype is preserved after 2 month of deletion, when we still observe crypt loss in the distal parts of the small intestine and the colon, whereas the proximal parts look almost normal (**new Supplementary Figure 1e**). This finding may explain why the mice do not show overt signs of disease. We believe that is not due to a replacement by ACC1-sufficient

cells, as we still see efficient ACC1-deletion in the epithelium after 2 month (in accordance with the literature. Doi:10.1002/gene.20042). We speculate that the differences in the severity of the pathology might at least partially reflect the presents of external sources of fatty acids, which may be available in higher concentrations in the proximal part of the small intestine. It is also possible that the microbiota in the proximal part of the colon can serve as a source of fatty acids. Nevertheless, we can also not completely rule out that some ISCs escape ACC1-deletion, as it has been for example reported recently by Berger et al. in mice with a conditional deletion of Hsp60 using the villin-cre^{ERT2} system (Doi:10.1038/ncomms13171). We included a discussion of these points to the revised version of the manuscript.

6. When does the crypt loss phenotype start in ACC1 KO? Do authors have data at early time point after TAM treatment and long-term observations for these ACC1 KO? Does crypt loss also occur in duodenum and jejunum, as authors only show the phenotype in ileum and colon in the figures? It is rather surprising that the animals can survive with this much crypt damage? Is the knockout partially efficient and the epithelium recovers over time with ACC1-WT cells? What cells remain proliferative to maintain villus epithelial turnover in the mutant?

Author response: Following the valuable suggestions of the reviewer, we have now included an extended analysis of the histological phenotype. Beside the results that we mentioned in our response above, we have analyzed the phenotype also at an earlier time point after tamoxifen injection (day5). At this time point, we did not observe any phenotypic changes (**new Supplementary Figure 1b**). This is consistent with the idea that genetic deletion and loss of the protein will take a few days. It is also possible that the presence of environmental fatty acids will slow down the process and the appearance of the phenotype.

7. The authors documented that quiescent Tert⁺ ISC subpopulation is less affected in ACC1 KO in vivo first, but then showed a significant alteration of Tert in vitro (Fig 3d). Explanation or discussion for these contradictory results in vivo vs in vitro is needed.

Author response: We thank the reviewer for pointing out this observation. Indeed we see a significantly reduced expression of tert in our in vitro experiments, while there is no clear reduction visible in vivo. Although we do not have a definite answer, we speculate that this difference may be explained by the presence of environmental factors in the in vivo situation. In general, the phenotype of ACC1-deficiency seems less pronounced in vivo, most likely due to the presence of fatty acids, which are available to a certain extent in the standard mouse chow (covering appr. 10% of energy) and which can be produced de novo in the liver and transported via the blood to the intestine. This may explain why we see also other differences between in vitro and in vivo, such as an significant increase in the expression of genes associated with secretory and absorptive enterocytes in vitro, but not in vivo. We have added a discussion of these points to the revised version of the manuscript.

8. The authors documented that they used 10 uM Soraphen A in the method section, but claimed they added 200 nM Soraphen A in figure legend of Fig. 3f. Could the authors clarify about that? What is the concentration of Soraphen A used in Fig 3e? Is it the same as the one used in Fig 3f? How do authors know the inhibition of organoid formation is due to inhibition of de novo FAS (Soraphen acts as an inhibitor) instead of the toxicity of Soraphen A itself? If the mice are treated with Soraphen A in vivo, will the mice show similar phenotype as ACC1 KO?

Author response: We apologize for the lack of clarity. We have used different concentrations of Sorafenib in human (10 μ M in Figure 3e of the original manuscript) and mouse organoid cultures (200 nM, Figure 3f of the original manuscript). We observed that a higher concentration of Sorafenib is necessary to observe the inhibitory effect in human organoids. We have tested different concentrations of Sorafenib in human organoid cultures and saw an effect starting around concentrations of 1 μ M (we now included this concentration in **the new Figure 3I**). This might reflect specific species differences in the IC50 or differences in the culture media between human and mouse (e.g. the presence of fatty acids. We have observed a similar effect previously, when treating mouse and human T cells. Doi:10.1038/nm.3704). In order to rule out that a possible off-target effect (or toxicity) of Sorafenib A may be responsible for the phenotype, we tested the compound 5-(tetradecyloxy)-2-furoic acid (TOFA) as another widely used specific inhibitor of ACCs. As **shown in the figure below**, addition of TOFA to WT organoid cultures induces the same effect on crypt domain formation as Sorafenib A or ACC1-deletion, indicating that the effect of SorA is not mediated by side effects, such as toxicity.

Figure legend: Crypts were isolated from ACC1^{lox/lox} mice and grown in organoid cultures. TOFA (Sigma; T6575, with the final concentrations as indicated in the figure) was added to the organoids at beginning of the culture. On day 5, organoids were imaged and the formation of crypt domains was quantified. >50 organoids were counted for each condition. Data was pooled from 3 independent experiments.

9. The authors analyzed gene expression in organoids instead of in vivo tissues, and claimed that few differences were found between ACC1 KO and controls. The organoid in vitro system is bringing more artificial growing conditions and organoids are growing in heterogeneity, when the in vivo tissues are available, gene expression analysis could be performed on in vivo tissues, such as the isolated ISCs, or crypts. The differences in organoid morphology in Fig. 3 suggest more differences should be observed in the RNAseq in Figure 4. Also, more than 2 replicates should be performed in RNAseq.

Author response: We thank the reviewer for this important comment and agree that a more in-depth transcriptomic analysis was necessary to improve the quality of our work. We therefore performed novel RNA-Seq experiments in triplicates to allow for a sound statistical analysis of the data. We appreciate the reviewer's suggestion to use ex vivo material. Nevertheless, we decided to perform our analysis on in vitro grown organoids. First, because we realized that the in vivo phenotype seems in general less pronounced, probably due to the presence of environmental factors, and depends (at least partially) also on the location.

We therefore believe that the in vitro conditions are easier to control, especially in terms of the availability of external fatty acids. According to our observations organoid growth in vitro is also better synchronized, which makes analysis possible at a specific time point of development and therefore allows for a clearer demarcation of specific differences. In fact, our new RNA-seq analysis confirms that ACC1-deficiency impacts critically on the expression of the ISC-related gene signature and genes associated with cell division and proliferation. We therefore believe that our approach faithfully reflects the functional impact of ACC1 and de novo FAS on ISC biology. The data of our novel analyses are shown in the **new Figures 3e-g, 4a** as well as **Supplementary Figures 3a and 5b**.

10. In Fig 4g, besides ACC1 KO, are we also expecting to see increased PPAR δ expression with GW501516 in controls (seems to be down in controls, GW vs DMSO)? It seems PPAR δ agonist also compromises stem cells in controls (Lgr5 expression in Fig 4f), why?

Author response: We thank the reviewer for pointing out this interesting observation. It is true that we do not observe an increase in nuclear amounts of PPAR δ in controls treated with GW501516 and a slight down regulation in Lgr5 expression. Consistent with these findings, we do also not observe functional differences between vehicle and GW501516 treated controls, such as numbers of crypt domain formation (**Figure 4e**) and secondary cloning capacity (**new Figure 4h**). It might be speculated that levels of PPAR δ activation are already at the upper threshold level in the controls under our culture conditions and agonist treatment would not lead to a significant additional increase in activation. Yet, although we cannot finally answer this question, we would like point out that this observation does not interfere with our major finding that GW501516 addition increases significantly nuclear PPAR δ protein levels in the ACC1-deficient organoids.

11. Can WNT agonists also rescue the ACC1 KO organoids?

Author response: Based on the reviewer's suggestion, we have tested whether the addition of the Wnt ligand Wnt3a can rescue the defect of ACC1-deficiency in our organoid system. As we show in the novel **Supplementary Figure 4d**, we tested several concentrations of Wnt3a, up to 100 ng/ml, but did not observe any rescue. Thus, excess addition of Wnt3a ligand alone is not sufficient to rescue the loss of de novo FAS. This indicates that de novo FAS may not directly influence the Wnt/ β -catenin pathway and that the products of de novo FAS are needed to sustain Wnt signaling via indirect mechanisms such as activation of PPAR δ . Together, this further corroborates the critical function of de novo FAS for the maintenance of ISC.

12. FAS in ACC1 KO vs controls should be an important evaluation in this study. The authors should design more experiments to investigate the function of ACC1 in FAS in the intestine. For example, authors could compare lipid profiles of ACC1 KO vs controls (in vivo samples, early time point after TAM, before crypt structure loss) and investigate fatty acid synthesis in ACC1 KO vs controls by incorporating labeled acetate in organoids.

Author response: We thank the reviewer for this helpful suggestion. As mentioned in our reply to the reviewers question above we have now performed metabolic ^{13}C flux assays to study FAS in our system. We have chosen this approach as it allows to directly follow the process of de novo FAS in a time dependent manner. We observed that in control organoids, glucose-derived ^{13}C is gradually incorporated in palmitate and its immediate saturated and unsaturated derivatives, such as stearate and oleate, indicating functional de novo FAS under normal WT conditions. In contrast, we did not observe significant ^{13}C incorporation in

all fatty acids analyzed, which indicates a complete shut-down of de novo FAS in the absence of ACC1 (**new Figure 4b**). Importantly, our results also confirm the lack of compensatory mechanism in the absence of ACC1 and underscores the importance of this enzyme for de novo FAS.

13. The C16 saturated fatty acid palmitate is the end product of de novo FAS, and the authors demonstrated that palmitate can rescue the phenotype of ACC1 KO. How about the effects of unsaturated fatty acid or other fatty acids (short chain, medium-chain, etc) in these ACC1 KO? Do ACC1 KO completely lose the function of de novo FAS? Can extra acetate (a precursor of acetyl-CoA) also rescue the phenotype of ACC1 KO?

Author response: We agree with the reviewer that it is of importance to test the effects of other fatty acids in our system. Our metabolic ¹³C flux analyses (**new Figure 4b**) confirm that de novo FAS is abrogated in ACC1-deficient organoids. While this phenotype can be rescued by the addition of palmitate, we observed that addition of acetate does not lead to a rescue (**new Supplementary Figure 4b**). This finding is in accordance with the idea that ACC1 is indeed essential for the de novo generation of fatty acids from the precursor acetyl-CoA. In addition, these results would argue against a lack of fatty acids as a source of energy (via FAO and OXPHOS) to explain the loss of ISC function upon ACC1 deficiency. Besides acetate, we have also assessed the effect of adding the unsaturated fatty acid oleate to ACC1-deficient organoid cultures. Although we tested different concentrations and time points of addition, we did not see any rescue in ACC1-deficient organoids (**see Figure below**). We do not have a definite explanation why palmitate, but not oleate can rescue the defect of de novo FAS in our in vitro system. Although this may hint towards a specific role of unsaturated fatty acids, there are many factors, including difference in uptake or kinetics of intra-cellular derivatization that could affect the different observations. We believe that an extensive analysis would be necessary to further clarify this issue. Yet, as our data clearly shows that not only addition of palmitate in vitro, but more importantly also treatment with a high fat diet in vivo can rescue the ACC1-deficient phenotype, we feel that this would go beyond the scope of the present manuscript.

Figure legend: Crypts were isolated from ACC1^{Δ/ΔIEC} mice and grown in organoid cultures +/- 4-OHT for 24h to induce ACC1-deletion. Oleate (Sigma, O1008, at final concentration of 100 μM, complexed with BSA) was added to the organoids after removing 4-OHT. On day 5, organoids were imaged and the formation of crypt domains was quantified.

More than or 30 organoids were counted for each condition. Data was pooled from 3 independent experiments. Other concentrations and time points of addition were tested (data not show).

Minor:

The ACC1 KO panel in Fig 1b seems to be cross-section tissues. It is hard to see the clear morphology of intestine in the KOs.

Author response: As mentioned above, we followed the valuable suggestions of the reviewer and performed a more extensive analysis of the histopathological changes in the different parts of the intestine. We have exchanged the pictures shown in **Figure 1b and c** accordingly.

In Fig 1c, any thoughts about why increased frequencies of innate leukocytes were only observed in the small intestine, but not in the colon?

Author response: Our histological analysis indicates that the structural changes in the colon of ACC1-deficient mice are more pronounced in the distal part, while the proximal parts appear less affected (**new Figure 1c**). Since for our immune phenotyping the cells were isolated from the lamina propria of the whole colon, we speculate that possible differences in the distal parts might have been diluted out.

The legend key is missed in Fig 2c, Fig 3d panel, although we can guess from other panels.

Author response: We apologize for the mistake. We added the legends in the revised version of the manuscript.

Is Fig 6b a representative figure? The tumor size is much bigger in ACC1 KO mice compared to controls. It does not fit for the distribution of tumor size panel, Fig 6e.

Author response: We are grateful to the reviewer for pointing out this important issue. We realized that the pictures we chose were not representative for our data. As the reviewer is stating correctly, we did not see differences in the tumor size distribution between WT and ACC1-deficient animals, but only in the total number of tumors. We have now included more representative pictures in the revised **Figure 6b**.

Reviewer 2

The authors characterize the intestinal phenotype in mice that lack ACC1, a protein that is involved in de novo fatty acid synthesis. The authors have previously explored the function of de novo fatty acid synthesis and ACC1 in immune T cell populations, while here the authors characterize the role of ACC1 within intestinal stem and progenitor populations, proliferating populations that need lipids for cellular division. Overall, we find the story interesting yet preliminary in its analysis and characterization of the phenotype, and, importantly, the tumor phenotype requires more clarification.

Specific comments:

1. Given that ACC1 function is essential and that this is first characterization of ACC1 loss-of-function in the intestine it would be informative to characterize the kinetics of the phenotype. How

early does the phenotype present? How long do adult intestinal KO animals live? Two weeks of deletion is an early timepoint and overt signs of disease may not have time to manifest. At longer timescales of excision, are disease pathologies present?

Author response: Following the valuable suggestions of the reviewer, we have included the analysis of the phenotype at different time points. Histological analysis at an early time point (one day after the last tamoxifen injection) did not reveal any pathological changes (**new Supplementary Figure 1b**). This is consistent with the idea that genetic deletion and loss of the protein will take a few days. As mentioned in our response to reviewer 1, we have now also included an extended analysis of the phenotype. We show that at day 14 the distal parts of the both colon and small intestine are stronger affected by the loss of ACC1, while in particular the proximal colon shows only a mild phenotype (**new Figure 1b and c**). This phenotype is preserved also at later time points (after 2 month of deletion), where we still observe crypt loss in the distal parts of the small intestine and the colon, whereas the proximal parts look almost normal (**new Supplementary Figure 1e**). This finding may explain why the mice do not show overt signs of disease. We believe that is not due to a replacement by ACC1 sufficient cells, as we still see efficient ACC1-deletion in the epithelium after 2 month. We speculate that the differences in the severity of the pathology might at least partially reflect the presence of external sources of fatty acids, which may be available in higher concentrations in the proximal part of the small intestine. It is also possible that the microbiota in the proximal part of the colon can serve as a source of fatty acids. We included a discussion of these points to the revised version of the manuscript.

2. Do you only achieve 90% excision efficiency because of epithelial escapers or are non-epithelial cells contaminating your sample?

Author response: We thank the reviewer for asking this interesting question. In general, we believe that the villin-cre-ERT2 is a very powerful system for the efficient deletion of genes in all epithelial cells, as it has been originally described by el Marjou et al. (Doi:10.1002/gene.20042) and which is also reflected by the high deletion efficiency in our system. As already mentioned above, deletion efficiency for ACC1 is still high (appr. 80-90%) in the small intestine after 2 month (**see Figure below**), in accordance with the original data by el Marjou et al, which indicates that the general rate of escape is not very high. It is thus possible that the lack of 100% deletion is reflecting a contamination with non-epithelial cells in our samples, as suggested by the reviewer. Nevertheless, we can also not completely rule that some ISCs escape ACC1-deletion, as it has been for example reported recently by Berger et al. in mice with a conditional deletion of Hsp60 using the villin-cre^{ERT2} system (Doi:10.1038/ncomms13171). We included a discussion of these points to the revised version of the manuscript.

Figure legend: RNA was extracted from ileum epithelial cells 2 month after the 1st tamoxifen treatment and qPCR was performed to determine ACC1 deletion efficiency. Data pooled from n=3 mice from one out of 2 experiments.

3. The epithelia appears severely damaged, please characterize further. Is there increased apoptosis or necrosis? Is the tissue still proliferating? Is there a change in the villi length, please quantify. Please quantify the number of crypts per a given length of intestine. The authors should better characterize the intestinal epithelial phenotype by quantitating the following: cell-death/apoptosis, proliferation, ISC fate mapping, in vivo ISC numbers, and survival (ie survival curve).

Author response: We appreciate the reviewer's suggestion to better characterize the phenotype in intestinal epithelial cells. First of all, we have now included the analysis of Lgr5-EGFP-IRES-Cre^{ERT2} x ACC1^{lox/lox} (ACC1^{Δ/ΔLgr5}) mice in our manuscript. Ex vivo analysis of tamoxifen treated ACC1^{Δ/ΔLgr5} mice showed a clear reduction of Lgr5⁺ ISCs, confirming a direct in vivo effect on Lgr5⁺ ISC upon ACC1 deletion (**new Figure 2e** and **new Supplementary Figure 2**). We have also performed novel RNA-Seq analysis of organoid cultures, which revealed significantly reduced gene signatures associated with cellular division and proliferation in the absence of ACC1 (**new Figure 3g** and **new Supplementary Figure 3a**). Although we did not observe significant changes in apoptosis, there was a clear increase in the frequency of dying/necrotic cells during the course of culture in ACC1-deficient organoids, indicative of enhanced cell death as a consequence of ACC1-deletion (**new Supplementary Figure 3b**). Accordingly, we observed a strongly reduced capacity of ACC1-deficient cultures to form organoids upon secondary cloning (**new Figure 3h**). Finally, we show now that crypts isolated from tamoxifen treated ACC1^{Δ/ΔIEC} mice display a greatly reduced capacity to form organoids in vitro (**new Figure 3i**). We feel that together, these novel data confirm a direct effect of ACC1-deficiency on ISC proliferation, survival and function.

4. The authors should also provide clarification regarding the metabolic adaptation that occurs when de novo FAS is dampened (ie compensation by ACC2, FASN) For instance, proliferating cells use lipids to divide so how do ISCs and progenitors adapt? Are the cells recycling lipids or are they cannibalizing lipid stores? Some evidence or discussion of these points would improve the manuscript.

Author response: We thank the reviewer for this helpful suggestion and agree that it is of importance to assess whether compensatory effects may occur in the absence of ACC1. We have now performed metabolic ¹³C flux analyses (**new Figure 4b**) which confirmed that de novo FAS is indeed abrogated in ACC1-deficient organoids. In addition, we found that addition of acetate does not lead to a rescue of the ACC1-deficient phenotype (**new Supplementary Figure 4b**). This finding is in accordance with the idea that ACC1 is indeed essential for the de novo generation of fatty acids from the precursor acetyl-CoA. It also indicates that there is no significant functional compensation for the lack of ACC1, e.g. by the activity of ACC2. In line with this, we did not observe major differences in the expression levels of both ACC2 and Fasn (**new Figure 4a**). It is possible that cells may preferentially use intracellular lipid stores when de novo FAS is blocked, which might explain the reduction in the levels of cellular neutral lipids (**Figure 4c**). The data that we provide in this manuscript demonstrate that cells (from both WT and ACC1-deficient organoids) can efficiently take up external fatty acids (**new Supplementary Figure 5a**), leading to a rescue of the ACC1-deficient phenotype when added in excess amounts both in vitro and in vivo. These results clearly suggest that addition of the end product of de novo FAS is sufficient to compensate for the lack of de novo FAS. As mentioned already above, our extended histological analysis revealed that the distal parts of both colon and small intestine are stronger affected by the loss of ACC1, while in particular the proximal

colon shows only a mild phenotype, which might at least partially reflect the presents of external sources of fatty acids in vivo. Regarding the implications on the metabolism, we would like to refer the reviewer to our response to question 10.

5. It is surprising that changes in intestinal cell types does not change. The MMP7 marker for Paneth cells looks more broadly expressed in ACC1-KO crypts compared to WT controls. Please quantify the histology. Also, the ACC1KO tissue for the PAS staining looks inconsistent with the MMP7, the crypts look very large and deep, hypertrophic.

Author response: As pointed out by the reviewer, we did not observe changes in the expression levels of genes associated with intestinal epithelial cell type in vivo, except for *Lgr5*. As suggested by the reviewer, we have quantified the presence of MMP7⁺ paneth cells as well as PAS⁺ goblet cells in the ileum part of the small intestine based on histological stainings. When counted on a cell per crypt basis, we indeed observed a small, but significant increase in the number of MMP7⁺ cells in ACC1-deficient mice (**new Figure 2d**). This finding would fit to our in vitro RNA-Seq data, which showed a shift towards the expression of IEC-associated genes in the absence of ACC1 (**new Figure 3f**). Yet, we did not see any changes when quantifying in the number of PAS⁺ goblet cells per crypt in histology (**new Figure 2d**). The differences between our in vitro and in vivo findings might be explained by a less pronounced in vivo phenotype as discussed above. As such, more subtle differences may only become significant in vivo in some specific approaches (e.g. in MMP7⁺ cell per crypt counting) but not others, such as more global RNA-expression analysis.

6. What is the frequency of *Lgr5*⁺ cells and *Olfm4*⁺ cells per crypt in different regions of the intestine? It is important to point out that *Olfm4* marks not only ISCs but early progenitors and is thus more broadly expressed compared to *Lgr5* in the crypt.

Author response: We thank the reviewer for raising this important question. As mentioned in our answer to the reviewer's question above, we have now included the analysis of ACC1^{Δ/Δ}*Lgr5*⁵ mice in our manuscript, in order better quantify the frequencies of *Lgr5*⁺ cell populations in vivo. Our analysis revealed significantly reduced frequencies of *Lgr5* high expressing ISCs in the lower part of the small intestine, as well as in the colon of tamoxifen treated *Lgr5*-ACC1-deficient mice. In crypts isolated from both locations we also observed a reduction in the frequencies of *Lgr5* intermediate expressing cells, which most likely represent direct progenitors of ISCs. In contrast, we did not see reduced frequencies of both *Lgr5* high and intermediate expressing cells in crypts isolated from the upper part of the small intestine. Thus our data using ACC1^{Δ/Δ}*Lgr5*⁵ confirm a specific loss of ISCs upon ACC1 deletion, but, in accordance with the histological findings, also reveal a less pronounced phenotype in the proximal part of the small intestine. The new data is included in the revised version of the manuscript in the novel **Figure 2e** and **new Supplementary Figure 2**.

7. In figure 3D, the levels of *Muc2*, *Lyz1*, and *Chgb* all appear to be increasing in the knock-out. Why is this the case? Is a loss of ACC1 pushing cells into a secretory, post-mitotic state in order to conserve lipids needed for division?

Author response: We thank the reviewer for bringing up this important point. Indeed, the analysis of our new RNA-Seq data confirms this initial observation and reveals increased expression of markers associated with the secretory lineage in the absence of ACC1 (**new Figure 3f**), and these findings are, at least partially,

reflected by enhanced numbers of MMP7⁺ Paneth cells per crypt in vivo (**new Figure 2d**). At the same time, we observe a strong downregulation of markers associated with stem cell identity and proliferation (**new Figure 3e and g**). It is therefore possible, that loss of ACC1 tips the balance of stem cell maintenance by proliferation towards differentiation into IEC lineages. The fact that ACC1-deletion results in a specific loss of stem cells may lead to a shift in the ratio between ISCs and other IEC types in the organoid cultures, which would add to the finding of increased IEC-associated transcripts in the absence of ACC1. We included a short discussion of this point to the revised version of the manuscript.

8. In their organoid assays, the authors delete ACC1 in propagating organoids in culture. However, it is also important to test stem cell function using the organoid assay by deleting ACC1 in vivo, prior to culturing, and then to quantify organoid clonogenicity as measure of stem cell activity. Further characterization of the organoids is also desired; for example, are the ACC1-null organoids proliferating and growing in size or what is the % live cells of organoids? The method the authors use allows organoids to form in the presence of residual ACC1 functional mRNA and protein. Crypt domains in organoids are an indication of differentiation; markers of differentiation increase in ACC1-null crypts, however, the crypt budding domains decrease. How do the authors explain this discrepancy?

Author response: Based on the reviewer's helpful suggestions we have performed several novel in vivo and in vitro experiments. As proposed, we have tested the ability of ex vivo isolated ACC1-deleted crypts to form organoids in vitro. The results of these clonogenicity assays demonstrate a clearly reduced capacity of ACC1-deficient crypt to grow into organoids (**new Figure 3i**), together confirming the importance of ACC1-mediated de novo FAS for ISC function. As mentioned in our reply above, transcriptional analysis indicated a strong loss of markers associated with cell cycle, DNA replication and proliferation upon loss of ACC1. Moreover, we observed significant increase of dying/dead cells in ACC1-deleted organoids during the course of organoid cultures. In accordance, ACC1-deficient organoids showed strongly reduced capacity to form novel organoids in secondary cloning assays (**new Figure 3h and Supplementary Figure 3b**). We believe that these findings confirm that lack of ACC1 results in loss of stem cell proliferation, function and maintenance. In our experiments shown in **Figure 3k**, we demonstrate that the initial growth of flow cytometry sorted Lgr5⁺ ISC into spheroids was severely impaired, when ACC1 was blocked by Sorafenib at the start of the culture, further confirming the direct impact of de novo FAS on ISC function. As pointed out correctly by the reviewer, we deleted ACC1 in most of our organoid assays at day one of culture, allowing organoids to grow into spheroids still in the presence of ACC1. In this case, we observe that loss of ACC1 manifests in reduced crypt domain formation. Notably, we observed the same phenotype when ACC1 was blocked by Sorafenib at a later time point, when sorted Lgr5⁺ cells already grew into spheroids (**Figure 3k**). We agree with the reviewer that crypt domain formation is an indication of differentiation. Nevertheless, the formation of crypt domain structures require the presence and continuous proliferation of ISCs at the tip of the growing crypt domain (as shown e.g. by Sato et al. Doi:10.1038/nature07935). We believe that in our case, the ISC that remain upon deletion of ACC1 may preferentially differentiate into IEC cell types, explaining the increase in transcripts associated with these cell types, but will lose their capacity to foster the growth of crypt domains due to their lack of self-maintenance.

9. The authors use a “growing medium” that contains WNT and RSPO. Why does this media not overcome the deficiency of b-catenin levels (WNT activation) as noted in Figure 4? A concentration

curve with increasing recombinant Wnt3a to test the role of Wnt signaling in rescuing the growth and budding defects would be helpful.

Author response: This is an important question that has also been raised by reviewer 1. We have performed additional experiments to assess the effect of Wnt3a reconstitution on ACC1-deficient organoids in our system. As shown in the **Supplementary Figure 4d**, we found that addition of Wnt3a (in several concentrations ranging from 25 μ M to 100 μ M) did not rescue the phenotype of ACC1-deficiency. Thus, excess addition of Wnt3a ligand alone is not sufficient to rescue the loss of de novo FAS. This indicates that de novo FAS may not directly influence the Wnt/ β -catenin pathway and that the products of de novo FAS are needed to sustain Wnt signaling via indirect mechanisms such as activation of PPAR δ . Together, this further corroborates the critical function of de novo FAS for the maintenance of ISC.

10. Why does PPARdelta activation rescue -what targets are activated? Does PPARdelta agonist rescue the in vivo phenotype? Is there an alternate route to making lipids other than through ACC1, such as compensation by ACC2 or FASN, or increased recycling of intracellular lipids? Can acetate rescue the ACC1-null organoids? In the organoid assays for GW/Palmitate/HFD, does less organoid death occur? Please quantify clonogenicity. Separately, PPARgamma is associated with lipogenesis, perhaps the authors could also check this PPAR family member. It would be informative to know how other PPAR family members (delta, alpha and gamma) are impacted by ACC1 loss. Can PPARgamma activation rescue the phenotype?

Author response: We appreciate the essential concerns that were raised also by the other reviewers. We have now performed several novel experiments in order to substantiate our findings. As mentioned in our response to the reviewer's earlier question, we have now performed metabolic ¹³C flux analyses which confirmed that there is no alternate route to synthesize lipids in the absence of ACC1 (**new Figure 4b**). This is further corroborated by our finding that acetate does not lead to a rescue (**new Supplementary Figure 4b**). We believe that this is in line with the idea that ACC1 is indeed essential for the de novo generation of fatty acids from the precursor acetyl-CoA. It also indicates that there is no significant functional compensation for the lack of ACC1, e.g. by the activity of ACC2 or Fasn. Of note, we do also not see significant changes in the expression levels of genes associated with de novo FAS (**new Figure 4a**). This is in accordance with findings by others, showing that interference with de novo FAS in neuronal stem cells by deletion of the regulator Spot14 (leading to reduced malonyl-CoA levels) did not result in major changes of FAS-associated genes (Doi:10.1038/nature11689). This indicates that the activity of FAS is less dependent on the transcriptional regulation of FAS genes, such as Fasn. We have furthermore assessed the survival of ACC1-deficient organoids treated with the PPAR δ agonist GW501516 and palmitate in secondary cloning assays. In both cases we observed a significantly increased ability to form secondary organoids, suggesting that activation of PPAR δ (as well as addition of palmitate) indeed enhances ISC function and maintenance (**new Figures 4h and 5d**). Although these data clearly indicate that activation of PPAR δ can rescue, at least partially, the defects in ISC upon ACC1-deletion, we did not observe a major impact on the expression levels of PPAR δ downstream genes, such as Cpt1a (**new Supplementary Figure 5b**). Cpt1a is an important rate limiting enzyme for mitochondrial fatty acid oxidation (FAO), as it mediates the transport of long- and intermediate-chain fatty acids into the mitochondria for beta-oxidation. In accordance with the findings by others (Doi: 10.1053/j.gastro.2019.11.031 and Doi: /10.1016/j.stem.2018.04.001), we observed that the blockade of FAO using the Cpt1a inhibitor Etomoxir reduces crypt domain formation in organoids, indicating an effect on ISC function. Importantly however,

we found that the addition of palmitate rescued crypt domain formation in ACC1-deficient organoids to levels seen in wild type controls, despite the presence of Etomoxir in the cultures (**new Figure 5e**). These findings indicate that metabolic adaptations, such as changes in FAO levels, do not play a major role in our system. This is in accordance with our findings that ACC1-deficient organoids do not show major defects in main metabolic functions, such as glycolysis and OXPHOS (**Supplementary Figure 4a**). Beyaz et al. have shown recently that PPAR δ activation by a high fat diet leads to increased stemness of ISCs, presumably by a direct interaction of PPAR δ with beta-Catenin (Doi:10.1038/nature17173). Moreover, a seminal report by Scholtysek et al. demonstrated a direct link between PPAR δ activation and stem cell function in osteoblasts (Doi:10.1038/nm.3146). Importantly, the authors found that activation of PPAR δ with GW501516 resulted in the accumulation of nuclear β -catenin in both osteoblasts and mesenchymal stem cells (which, notably, was not observed upon activation of PPAR γ). Similarly, activation of PPAR δ induced the Wnt-dependent mRNA expression of osteoblast marker genes. This is in accordance with our findings that PPAR δ activation can rescue the defect of ACC1-deficiency, leading to increased nuclear β -catenin accumulation and ISC-function (**Figures 4e-h**). We therefore believe that activation of PPAR δ rather promotes ISC function in our system by stabilizing β -catenin function (as indicated by the increase of nuclear β -catenin in our rescue experiments). Yet, as we have discussed in our manuscript, we appreciate that also several other mechanism might be of importance to explain the dependence of ISCs on de novo FAS, as we have e.g. also observed changes in the intracellular neutral lipid levels upon loss of ACC1. We nevertheless respectfully feel that a more detailed analysis of all mechanistic implications would go beyond the scope of the present manuscript. We accordingly extended our discussion on possible down-stream effects upon loss of de novo FAS and adjusted our statements regarding the impact of PPAR δ in the revised version of the manuscript to account for this fact.

11. What are the genes in the heat map? Please annotate. Why does it appear that the ACC1-WT genes flip in expression from day1 to day4? Please clarify the headings in the supplemental table, Currently, it is unclear what the column headings indicate. Also, in general Ppar levels change very little upon stimulation. A better readout of PPARdelta activity is its transcriptional targets.

Author response: We agree with the reviewer that the presentation of the transcriptomic data was not optimal in the original manuscript. As similar questions were raised also by other reviewers, we decided to perform novel RNA-sequencing (instead of microarray) and, to allow for a sound statistical analysis of the data, the transcriptomic analysis was this time performed in triplicates. Moreover, we concentrated our analysis for the revised version of the manuscript on the comparison between WT and ACC1-deficient organoids on day 4 upon tamoxifen treatment, in order to better define the consequences of ACC1-deletion on the transcriptional level. In fact, direct comparison of WT and ACC1-deleted organoids confirms that ACC1-deficiency impact critically on the expression of the ISC-related gene signature (**new Figure 3e**). We therefore believe that our approach faithfully reflects the functional impact of ACC1 and de novo FAS on ISC biology. As mentioned in our discussion to the reviewers' questions above, the novel RNA-Seq data also enabled us to better define the consequences of ACC1-deficiency on cellular division and proliferation as well as on the metabolic implications. The novel data derived from the RNA-seq are shown in the **new Figures 3e-g, 4a and Supplementary Figure 3a and 5b**.

12. The tumor data is not very strong. The ACC1 has larger fractions of bigger tumors but fewer total tumors indicating that perhaps tumors are not clonal and have fused. These data require more analysis and perhaps assessing tumor burden at earlier time points would help.

Author response: We appreciate the essential concerns that were raised by the reviewer and apologize for the lack of clarity in the presentation of our results. We realized that the data provided in the original manuscript was not suited to exemplify the differences in the tumor formation in WT and ACC1-deficient mice. We have updated the representative images shown in **Figure 6b** accordingly. Regarding the differences in tumors size, we would like to allude to the fact that we did not see significant differences between WT and ACC1-deficient mice. We have adapted **Figure 6d** and included error bars as well as p-values in order to make this point more clear. In addition, we have extended our analysis and included histological scoring at the end point of our experiments. As shown in new **Figure 6e**, we find increased inflammatory pathology in ACC1-deficient mice, in accordance with the decrease colon length observed in these animals. The finding that ACC1-deficient mice show rather enhanced inflammation-related pathology both in the acute phase of DSS treatment (**Supplementary Figure 6**) as well as at the end point of the experiment is of importance, since it demonstrates that reduction in tumor numbers is not a simple consequence of reduced inflammation in those mice. Together we respectfully believe that our results clearly indicate that ACC1-deficiency interferes with the onset of tumor formation. Yet, when initial formation of adenomas has taken place, they further progress and develop into tumors to the same extent in ACC1-deficient mice.

Minor:

Consider normalizing the QPCR data to control.

Author response: We thank the review for this suggestion. We have accordingly adjusted the presentation of qPCR data in all Figures to show relative expression compared to the control.

Sup fig 3a schematic is not consistent with Fig6a.

Author response: We apologize for the lack of clarity. We have changed the description in the manuscript and in the figure legends in order to make it clear that the histopathological analysis shown in **Supplementary Figure 6** was performed during the acute phase of colitis directly after the first cycle of DSS treatment.

Figure 3D needs significance calculated

Author response: Figure 3d has been reorganized in the revised version of the manuscript and significance has been calculated.

Error bars are needed on Supp Fig 1

Author response: We apologize for this mistake and have added error bars to the figure.

Clarification is needed in lines 136-138 as to which cell type is being measured.

Author response: We thank the reviewer for pointing out this mistake. We have corrected the sentence to make clear that intestinal epithelial cells (IEC) have been analyzed.

Reviewer 3

In this manuscript, the authors investigate the role of ACC1 in the fatty acid synthesis pathway in maintaining Lgr5+ intestinal stem cell function. While these findings are interesting and worth exploring, critical conclusions from this study are not sufficiently supported by the presented data. More detailed comments are as follows:

Major comments:

1. In the de novo fatty acid synthesis pathway, ACC1 mediates acetyl-CoA conversion to malonyl-CoA, then FAS mediates the malonyl-CoA to generate the end product Palmitate. Inhibition or deletion of ACC1 blocks the fatty acid synthesis pathway. Does extra acetyl-CoA drive beta-oxidation in all the models, therefore driving the intestinal stem cell proliferation? In addition, malonyl-CoA also plays an important role in the regulation of fatty acid oxidation in the mitochondria as an inhibitor of carnitine palmitoyl transferase 1, which performs the first step in the transfer of long-chain fatty acyl CoA into mitochondria for their oxidation. Although the authors generated intestinal organoids from ACC1-deleted IEC mice and observed only minor effects on OCR and no critical impact on mitochondrial metabolism, why are the basal OCR levels of ACC1 IEC KO mice higher than the ACC1 flox mice? Are there any controls (cell number or protein amount) for the Seahorse organoid assay? Is this experiment carried out on day 1 or day 4?

Author response: We thank the reviewer for raising this important point and agree that a better understanding of the role of FAO in our system is necessary. We have now performed several novel experiments in order to clarify this question. First of all, we directly studied the impact of ACC1-deletion on the de novo synthesis of fatty acids using metabolic ¹³C flux assays. The results confirm that de novo FAS has been completely blocked upon ACC1 deletion, ruling out alternative or compensatory routes of FAS in IECs (**new Figure 4b**). As stated correctly by the reviewer, an important role of FAO for the maintenance of ISC has been reported recently by others (Doi: 10.1053/j.gastro.2019.11.031 and Doi: /10.1016/j.stem.2018.04.001). It is therefore possible that interference with de novo FAS may impact ISC function by affecting FAO via direct or indirect mechanisms. We have tested the effect of acetate-addition, as a precursor for acetyl-CoA, in our system. However, we found that the addition of acetate was not sufficient to rescue the effect of ACC1-deletion (**new Supplementary Figure 4b**), indicating that providing a substrate for FAO and OXPHOS is not sufficient to drive ISC function in the absence of de novo FAS. We have also tested the effect of Cpt1 inhibition in our system using Etomoxir as an inhibitor of this enzyme. Of note, we have taken care to use a low concentration of Etomoxir to ensure for a specific inhibitory effect on Cpt1, without the side effect that has been described by us and others on enzymes in the respiratory chain (DOI: 10.1111/imr.12655). Indeed, we found that blocking Cpt1 reduced crypt domain formation in wild type organoids, suggesting an effect on ISCs and thus confirming the findings by Chen et al and Mihalyova et al. (Doi: 10.1053/j.gastro.2019.11.031 and Doi: /10.1016/j.stem.2018.04.001). Importantly however, our results further demonstrate that the addition of palmitate rescued crypt domain formation in ACC1-deficient organoids to levels seen in wild type controls, despite the presence of Etomoxir in the cultures (**new Figure 5e**). We believe that these findings indicate that fatty acids, produced by de novo FAS or when added to the cultures, do not drive ISC function by stimulation FAO. Regarding the presentation of the Seahorse data, we apologize for the lack of clarity in the experimental description. Analysis has been performed on day 3 of organoid culture (24h of 4OHT treatment followed by another 48h of culture). We have corrected the Figure legend accordingly. We completely agree with the reviewer that it is of critical importance to carefully establish and control the organoid system for the Seahorse application. As such, we have tested

several methods for normalization of the data (e.g. cell number or amount of protein). In our hands, normalization of the data to the number of viable organoids, counted by microscopy in the respective well, has proven to provide the most reliable and reproducible results. As the reviewer has pointed out, we observed a certain increase in basal OCR levels in the ACC1-deficient organoids. We therefore cannot rule out compensatory mechanisms that lead to enhanced OXPHOS upon ACC1-deficiency. We have added a short discussion in our manuscript to acknowledge this point. Yet, taken together our novel findings described above, we respectfully believe that the phenotype of ACC1-deficiency that we describe in our study cannot be explained mainly by an effect on FAO.

2. The authors found increased numbers of innate leukocytes such as DCs and Macrophages in the small intestine but not in the colon of ACC1 KO mice by flow cytometry in Figure 1b. ACC1 KO mice were also challenged with AOM-DSS and the authors concluded that inhibition of ACC1 reduced the formation of tumors. While the tumor load decreased in Figure 4c, both Figure 4b and 4e show that tumor sizes increased in ACC1 IEC KO mice. In addition, reduced colon lengths in Figure 4b and 4d both indicated more severe inflammation in ACC1 IEC KO mice. Does ACC1 contribute to tumor progression or not? All other data presented are ACC1 deletion in healthy states, such as ACC1-mediated PPARdelta/b-catenin activation in ISCs.

Author response: This is an important issue that has also been raised by reviewer 2. As mentioned in our response to reviewer 2 above, we realized that earlier data provided was not optimal to exemplify the differences in the tumor formation in WT and ACC1-deficient mice. We have therefore updated the representative pictures shown in **Figure 6b** to better reflect the difference in tumor formation that were observed between the two genotypes. Regarding the differences in tumors size, we would like to apologize for the lack of clarity in the presentation of the results. In fact, we did not see significant differences between WT and ACC1-deficient mice. We have adapted **Figure 6d** in the revised version of the manuscript and included error bars as well as and p-values in order to make this point more clear. As the reviewer stated correctly, our data points towards increased inflammation in the colons of ACC1-deficient mice in the AOM/DSS model. To better characterize this phenotype, we have extended our analysis and included histological scoring at the end point of our experiments. As shown in **new Figure 6e**, we find increased inflammatory pathology in ACC1-deficient mice, in accordance with the decrease colon length observed in these animals. The finding that ACC1-deficient mice show rather enhanced inflammation-related pathology both in the acute phase of DSS treatment (**Supplementary Figure 6**) as well as at the endpoint of the experiment is of importance, since it demonstrates that reduction of tumors is not a simple consequence of reduced inflammation in those mice. Together we respectfully believe that our results clearly indicate that ACC1-deficiency interferes with the onset of tumor formation. Yet, when initial formation of adenomas has taken place, they further progress and develop into tumors to the same extent in ACC1-deficient mice.

Detailed comments:

1. Figure 1a, significance should be indicated.

Author response: Significances have been added to **Figure 1a** in the revised version of the manuscript.

2. Figure 1b, the small intestine section of ACC1 KO IEC mice is cut at an angle and therefore cannot be used to count villi length or crypt shapes. Scale bars are missing.

Author response: Following the valuable suggestions of the reviewers, we have now extended the characterization of the histological phenotype in the revised version of the manuscript. The pictures of the sections shown in the **new Figure 1b** have been adapted accordingly. We also adjusted the scales bars in the pictures to make them more visible. As we have mentioned in our responses to the other reviewers, our extended analysis revealed that the distal parts of the both colon and small intestine were stronger affected by the loss of ACC1, while in particular the proximal colon showed a milder phenotype (**new Figure 1b and c**). This phenotype was preserved after 2 month of deletion, where we still observed crypt loss in the distal parts of the small intestine and the colon (**new Supplementary Figure 1e**), whereas the proximal parts appeared almost normal (which may explain why the mice do not show overt signs of disease). We believe that this not due to a replacement by ACC1 sufficient cells, as we still see efficient deletion in the epithelium after 2 month (in accordance with the literature, Doi:10.1002/gene.20042). We believe that this might at least partially reflect the presents of external sources of fatty acids, which may be available in higher concentrations in the proximal part of the small intestine. It is also possible that the microbiota in the proximal part of the colon can serve as a source of fatty acids. We included a discussion on these points to the manuscript.

3. Figure 1c, the Y axis goes directly from 10 to 100. It is difficult to determine the T, B and Macrophage percentages from the graph. The gating strategy could be put into Supplementary data, and real flow images with X and Y axis comparing ACC1^{flx} and ACC1^{KO} mice should be presented in the main figure.

Author response: We thank the reviewer for these suggestions. We have changes the presentation of the immune cell percentages to a linear scale. The gating strategy was put into the supplements (**Supplementary Figure 1c**) and representative flow cytometry plots comparing WT and ACC1-deficient mice have been added to the main Figure (**new Figure 1d**).

4. Figure 2, since authors have Lgr5-EGFP mice, have they examined GFP⁺ cells in the ACC1 KO mice? How about EdU incorporation in ACC1 KO mice?

Author response: We agree with the reviewer that this is an important missing experiment. As we have mentioned also in our responses to the other reviewers, we have now included the analysis of Lgr5-EGFP-IRES-Cre^{ERT2} x ACC1^{lox/lox} (ACC1^{Δ/ΔLgr5}) mice in our manuscript, in order better quantify the frequencies of Lgr5⁺ cell populations in vivo. Our analysis revealed significantly reduced frequencies of Lgr5 high expressing ISCs in the lower part of the small intestine, as well as in the colon of tamoxifen treated ACC1^{Δ/ΔLgr5} mice. In crypts isolated from both locations we also observed a reduction in the frequencies of Lgr5 intermediate expressing cells, which most likely represent direct progenitors of ISCs. In contrast, we did not see reduced frequencies of both Lgr5 high and intermediate expressing cells in crypts isolated from the upper part of the small intestine. Thus, our data using ACC1^{Δ/ΔLgr5} confirm a specific loss of ISCs upon ACC1 deletion, but, in accordance with the histological findings presented in **the new Figure 1b**, also reveal a less pronounced phenotype in the proximal part of the small intestine. The new data is included in the revised version of the manuscript in the novel **Figures 2e and Supplementary Figure 2**. Although we have not directly measured proliferation in vivo, we have extended our analysis using organoids cultures in vitro. Analysis of new RNA-Sequencing data revealed significantly reduced gene signatures associated with cellular division and proliferation in the absence of ACC1 (**new Figure 3g and Supplementary Figure 3a**).

Although we did not observe significant changes in apoptosis, there was a clear increase in the frequency of dying/necrotic cells during the course of culture in ACC1-deficient organoids, indicative of enhanced cell death as a consequence of ACC1-deletion (**new Supplementary Figure 3b**). Accordingly, we observed a strongly reduced capacity of ACC1-deficient cultures to form organoids upon secondary cloning (**new Figure 3h**). Finally, we show now that crypts isolated from tamoxifen treated ACC1^{ΔIEC} mice display a greatly reduced capacity to form organoids in vitro (**new Figure 3i**), together confirming a direct effect of ACC1-deficiency on ISC proliferation, survival and function.

5. Figure 2c, why is the Y axis so low, relative expression to which gene? Any explanation why Tert gene or quiescent subpopulation of ISCs does not change in ACC1 KO mice? Does it also not change in AOMDSS injury model?

Author response: We thank the reviewer for pointing out this observation. In general we normalized all expression data to the expression of the house keeping gene beta-actin. As beta-actin is expressed to a high level (usually at CT values < 15 in our hands) expression levels of the genes shown in Figure 2c appear relatively low. Following the suggestion of reviewer 2, we have decided to show qPCR-derived gene expression data in our revised manuscript as fold changes compared to control, to better point out differences in gene expression between analyzed groups. Indeed we see a significantly reduced expression of tert in our in vitro experiments, while there is no clear reduction visible in vivo. As mentioned above, we believe that this difference could be explained by the presence of environmental factors in the in vivo situation. In general, the in vivo phenotype seems less pronounced in vivo, most likely due to the presence of fatty acids, which are present to a certain extent in the standard mouse chow and which can be produced de novo in the liver and transported via the blood to the intestine. We have added a discussion of these findings to our revised version of the manuscript

6. Figure 3b, when you knockout ACC1, what happens to FAS, ACC2 expression?

Author response: We agree that a more direct analysis of the impact of ACC1-deficiency on de novo FAS is necessary in our system. To do so, we have performed 13C based flux assays as a direct readout for the cellular de novo synthesis of fatty acids. While we observed a gradual increase of 13C in palmitate and its derivatives, our results show clearly that FAS is disrupted in ACC1-deficient organoids (**new Figure 4b**). Together, these results confirm that ACC1 is indispensable for de novo FAS in IECs. In addition, this finding also indicates that there is no significant functional compensation for the lack of ACC1, e.g. by the activity of ACC2. Of note, we do also not see significant changes in the expression levels of genes associated with de novo FAS (**new Figure 4a**). This is in accordance with findings by others, showing that interference with de novo FAS in neuronal stem cells by deletion of the regulator Spot14 (leading to reduced malonyl-CoA levels) did not result in major changes of FAS-associated genes (Doi:10.1038/nature11689). This suggests that the activity of FAS is less dependent on the transcriptional regulation of FAS genes, such as Fasn.

7. Although in Figure 4 the authors claimed ACC1 contributions to ISC function is dependent on PPARdelta/b-catenin pathway, no PPARdelta pathway genes such as Cpt1a, Angptl4, Pdk1, Hmgs2 etc., show significant expression changes (Supp table 1). Please explain this discrepancy.

Author response: We appreciate that this is an important open question. We performed additional experiments in order to substantiate the role of PPAR δ in our system. As shown in the **new Figure 4h**, we

have assessed the survival of ACC1-deficient organoids treated with the PPAR δ agonist GW501516 in secondary cloning assays. We observed a significantly increased ability to form secondary organoids, suggesting that activation of PPAR δ indeed enhances ISC function and maintenance. Together with our initial results, these data clearly indicate that activation of PPAR δ can rescue, at least partially, the defects in ISCs upon ACC1-deletion. As the reviewer pointed out correctly, we did not observe a major impact on the expression levels of classical PPAR δ downstream genes, such as Cpt1a or genes involved in FAO, a finding that we confirmed using novel RNA-Seq analysis (**new Supplementary Figure 5b**). We believe that this result is in accordance with our findings that ACC1-deficient organoids do not show major defects in main metabolic functions, such as glycolysis and OXPHOS (**Supplementary Figure 4a**). Along this line, as mentioned also in our responses to the other reviewers, we observed that ACC1-deficiency can be rescued when Cpt1 is blocked by Etomoxir and therefore independently of FAO (**new Figure 5e**). Together, these findings indicate that metabolic adaptations, such as changes in FAO levels, do not play a major role in our system. As we discuss in more detail to the reviewer's question 10 below, several groups have shown convincingly that PPAR δ can physically bind to beta-catenin (Doi:10.1038/nature17173 and Doi:10.1038/nm.3146). We believe that activation of PPAR δ rather promotes ISC function in our system by stabilizing β -catenin function (as indicated by the increase of nuclear beta-catenin in our rescue experiments (**Figure 4g**)). Yet, as we have discussed in our manuscript, we appreciate that also several other mechanism might be of importance to explain the dependence of ISCs on de novo FAS, as we have e.g. also observed changes in the intracellular neutral lipid levels upon loss of ACC1. We nevertheless respectfully feel that a more detailed analysis of all mechanistic implications would go beyond the scope of the present manuscript. We accordingly extended our discussion on possible down-stream effects upon loss of de novo FAS and adjusted our statements regarding the impact of PPAR δ in the revised version of our manuscript to account for this fact.

8. Figure 3b, while the expression of Lgr5 and Tert are not increased in ACC1KO organoids from day 1 to day 5, the expression of Muc2, Lyz1 and Chgb appears to gradually increase in ACC1KO organoids from day 1 to day 5. Please explain this discrepancy.

Author response: Indeed, the analysis of our new RNA-Seq data confirms the initial observation and revealed increased expression of markers associated with the secretory lineage in the absence of ACC1 at day for of organoid culture (**new Figure 3f**). These findings are, at least partially, reflected by enhanced numbers of MMP7⁺ Paneth cells per crypt in vivo (**new Figure 2d**). At the same time, we observed a strong decrease of markers associated with stem cell identity and proliferation in ACC1-deficient organoids (**new Figures 3e and g**). As we have mentioned in our reply to a related question by reviewer2, we believe that this may indicate that loss of ACC1 tips the balance of stem cell maintenance by proliferation towards differentiation into IEC lineages. The fact that ACC1-deletion results in a specific loss of stem cells may also lead to a shift in the ratio between ISCs and other IEC types in the organoid cultures, which would add to the finding of increased IEC-associated transcripts in the absence of ACC1. We included a shorts discussion of this point to the revised version of our manuscript.

9. Figure 3f: The scale bar is missing. Is the SorA-treated organoid placed in growing medium at day 1? Would be helpful to include both low and high magnification images to show the sizes of each one.

Author response: We apologize for the lack of clarity in the description and presentation of these data (**Figure 3k** in the revised manuscript). We specified our description in the methods section to indicate that

“Soraphen A... was added into organoid cultures ... either at the beginning of the culture in growing medium directly after embedding sorted ISCs in matrigel, or directly after changing to differentiation medium.” We have included side-by-side images of organoids from both SorA treated and control organoids at the same magnification, which we believe makes it easier to appreciate the smaller size of the organoids grown in the presence of SorA. To enhance the clarity, we have adjusted the color and thickness of the scale bars in the images.

10. Figure 4d, again GW1516 activates PPARdelta to elicit the beta-oxidation pathway. Therefore, one cannot conclude from this data that ACC1 contribution to ISC maintenance is FAS-dependent.

Author response: We agree with the reviewer that activation of FAO and the genes associated with this pathway is a well-documented effect of PPAR δ activation. Yet, as discussed above, we do not see a major impact on the transcription of genes implicated in FAO when comparing the transcriptome of WT and ACC1-deficient organoids (**new Supplementary Figure 5b**), consistent with the finding described above that rescue of the phenotype is possible despite blocking FAO (**new Figure 5e**). In addition to the effect on fatty acid metabolism, PPAR δ or its direct transcriptional targets have been demonstrated to influence critically on a variety of cellular functions, such as cellular differentiation, metabolism, as well as inflammation and stem cell maintenance (Doi:10.1158/1078-0432.CCR-16-0775). As mentioned above, Beyaz et al. have shown recently that PPAR δ activation by a high fat diet leads to increased stemness of ISCs, presumably by a direct interaction of PPAR δ with β -catenin (Doi:10.1038/nature17173). Moreover, a seminal report by Scholtyssek et al demonstrated a direct link between PPAR δ activation and stem cell function in osteoblasts (Doi:10.1038/nm.3146). Importantly, the authors found that activation of PPAR δ with GW501516 resulted in the accumulation of nuclear β -catenin in both osteoblasts and mesenchymal stem cells (which, notably, was not observed upon activation of PPAR γ). Similarly, activation of PPAR δ induced the Wnt-dependent mRNA expression of osteoblast marker genes. This is in accordance with our findings that PPAR δ activation can rescue the defect of ACC1-deficiency, leading to increased nuclear β -catenin accumulation and enhanced function, maintenance and marker expression of ISCs (**Figures 4e-h**). In this respect, we believe that intracellular de novo FAS may be important for the production of internal lipid ligands of PPAR δ (as has been discussed by Beyaz and Yilmaz Doi:10.1158/1078-0432.CCR-16-0775) and therefore could explain the dependency of ISC on de novo FAS. Of note, we found that activation of PPAR δ did not lead to a complete rescue of the ACC1-deficient phenotype. As discussed above, this may point towards other important functions of de novo FAS for ISC. We have adjusted our statements on the dependency on PPAR δ activation and have extended our discussion on the role of PPAR δ and other possible down-stream effects upon loss of de novo FAS in the revised version of the manuscript.

11. A similar question with palmitate and HFD compensation. What happens to the OCR profile when palmitate is added to ACC1 KO organoids?

Author response: We would like to answer this question by mainly referring to our discussion of the reviewer's question above. Since we believe that our findings rather speak against a prominent role of FAO in our system, we think that the rescue of the ACC1-deficient phenotype by palmitate is explained, at least partially, by providing ligands for PPAR δ activation (as evidenced by increased nuclear β -catenin accumulation, ISC maintenance and marker expression (**Figs 4e-h**)). Again, we cannot rule out that other mechanism may play an important role, which we have discussed now more extensively in the revised version of the manuscript.

12. Figure 5. As mentioned above, ACC1 KO mice upon AOMDSS challenge seem to have severe inflammation and increased tumor burden rather than reduction of tumorigenesis.

Author response: We thank the reviewer for pointing out this observation. We would like to refer to the second question of the reviewer (major comments), as we have discussed these points in detail in our response to this question.

REVIEWER COMMENTS

Reviewer #1 (Remarks to the Author):

Additions to metabolomics and increased rigor with the RNAseq replicates are certainly an advance from the previous submission. However, the mechanism through which ACC1 loss leads to collapse of stem cells, and why the phenotype is highly regional to the distal ileum and colon are not very well explained. This reviewer feels that the manuscript is highly interesting as a phenotype, but remains incompletely understood mechanistically.

Reviewer #2 (Remarks to the Author):

We thank the authors for addressing many of our concerns. The added data, clarifications, and formatting changes improve the overall message of the manuscript describing how de novo fatty acid synthesis affects ISCs and the intestine. We have a few remaining concerns:

Much of the data depend upon in vitro administration of 4-OH tamoxifen. Can the authors confirm that the concentration used does not have a detrimental effect on genotypic Vil-CreERT2 organoids? The manuscript methods indicate that the working concentration is 300nM. This is high enough to cause concern that the activated metabolite may be inducing undesired effects. Additionally, activated Vil-CreERT2 has shown to have effects on stem cells in vivo (PMID: 30449703). It would be important to know how much of the characterized phenotypes are due to the loss of ACC1 or simply the effect of activated Cre and/or a high dose of 4-OH tamoxifen.

We would like more clarity on the model. Multiple lines of evidence suggest exogenous Wnt3a in culture should rescue the ACC1-KO phenotype: 1) beta-catenin is lowered in the mutant organoids and Wnt3a would stimulate more beta-catenin, 2) a HFD rescues the in vivo ACC1-KO histological phenotype in which the diet has been shown to increase Wnt signaling in ISCs. Can PPAR δ agonist rescue the in vivo phenotype? Or alternatively, does the loss of Ppard recapitulate this phenotype in vitro? The rescue experiment is very valuable here and your data are surprising. Regarding SuppFigD, does the addition of Wnt3a to control organoids have an effect, that is, is your batch of Wnt3a adequately functional. The results dealing with Wnt3a and ACC1-KO are unexpected and require additional considerations.

In the Discussion (lines 349-351) "Our findings here support a model in which activation of PPAR δ is critical to sustain normal ISC function also under physiological conditions." We find this statement to be an overinterpretation. Beyaz et al. use the organoid assay to show that Ppard loss had no effect on organoid growth, indicating it is not critical for ISC function under physiological conditions. Without further loss of function studies, this statement should be omitted or rephrased.

Minor.

-In organoid assays, it would be informative to know if the crypts were harvested from the proximal, distal, or total small intestine. Please indicate in the text or figure legends.

-Supp Fig1E. Please confirm the duodenum and ileum images are properly assigned. They appear swapped.

-Supp Fig2 - Please clarify the label on the progenitor population "%Lgr5^{high}"

-Line 205. The sentence is unclear.

Reviewer #3 (Remarks to the Author):

On the positive side, the authors addressed the majority of the questions. On the negative side, some of the answers and conclusions are confusing.

Comments:

1. As mentioned by Reviewer 1, the phenotype of the ACC1 IEC KO mice does not consistently show the damage of crypt structure and appears to have limited impact on other cell types. For example, in Figure 1c, both proximal and distal colon show decreased goblet cells in the H&E image, as does the MMP7 staining in Figure 2d. However, in both Figures 2c and 2d, PAS staining shows no change in differentiated cell types. Moreover, in Sup Figure 1b, there appear to be more goblet cells in the ACC1 IEC KO colon. Perhaps there's an explanation. Bottomline this needs to be clarified.
2. In Figure 2e, how many crypts were isolated from Lgr5-EGFP WT and ACC cKO mice to do the sorting? For the flow map presented in Fig 2e, it's unclear how many crypts and cells were used which impacts statistical relevance.
3. Following question 1, in Fig 3f, the RNAseq data of WT and ACC IEC KO mice also showed significant differences of marker genes of differentiated cell types. Do the authors have any explanation for the differences?
4. Does GSEA of RNAseq data call out changes in fatty acid pathway genes?
5. In the AOM-DSS model, are tumors per colon representing one colon segment in each group or the average tumor numbers per mice? Please indicate the "n" number. Also please present more than one image per group. There also appear to be tumors in the ACC lox mice, but it is unclear whether they are not labeled or not counted or both. Also, because ACC1-deficient mice showed increased inflammation but reduction of tumors, I am unclear as to the authors' explanation "The finding that ACC1-deficient mice show rather enhanced inflammation-related pathology both in the acute phase of DSS treatment as well as at the endpoint of the experiment is of importance, since it demonstrates that reduction of tumors is not a simple consequence of reduced inflammation in those mice."
6. Although the authors explained the discrepancy of PPARdelta downstream genes (Cpt1a, Angptl4, Pdk1, Hmgcs2, etc) having no significant changes by proposing another role in promoting ISC function by stabilizing Beta-catenin function, we agree that further future studies are needed for delineating these two pathways.

We appreciate the constructive critiques by the reviewers that have been very useful to further improve the quality of our work. We have now followed each of the major concerns and performed all experiments suggested.

We would like to address the referees' questions as follows:

Reviewer #1 (Remarks to the Author):

Additions to metabolomics and increased rigor with the RNAseq replicates are certainly an advance from the previous submission. However, the mechanism through which ACC1 loss leads to collapse of stem cells, and why the phenotype is highly regional to the distal ileum and colon are not very well explained. This reviewer feels that the manuscript is highly interesting as a phenotype, but remains incompletely understood mechanistically.

Author's response: We thank the reviewer for the careful and critical evaluation of our study. We are grateful for the reviewers' comments and suggestions, which have helped us to increase the quality and impact of the study. We believe that we present an important insight into the molecular consequences of ACC1-mediated loss of de novo FAS in ISCs. In the revised version of the manuscript, we included additional work demonstrating that PPAR δ activation can rescue the phenotype of ACC1-deletion also in vivo, further stressing the critical impact of this pathway (**new Figure 4e**). We agree with the reviewer that there are question remaining, but respectfully feel that this would go beyond the scope of the present study. We hope that the reviewer still finds that the manuscript in its present form is suitable for publication.

Reviewer #2 (Remarks to the Author): We thank the authors for addressing many of our concerns. The added data, clarifications, and formatting changes improve the overall message of the manuscript describing how de novo fatty acid synthesis affects ISCs and the intestine. We have a few remaining concerns:

Q1: Much of the data depend upon in vitro administration of 4-OH tamoxifen. Can the authors confirm that the concentration used does not have a detrimental effect on genotypic Vil-CreERT2 organoids? The manuscript methods indicate that the working concentration is 300nM. This is high enough to cause concern that the activated metabolite may be inducing undesired effects. Additionally, activated Vil-CreERT2 has shown to have effects on stem cells in vivo (PMID: 30449703). It would be important to know how much of the characterized phenotypes are due to the loss of ACC1 or simply the effect of activated Cre and/or a high dose of 4-OH tamoxifen.

Author's response: We thank the reviewer for pointing out this interesting publication. In order to directly test for an ACC1-independent effect of tamoxifen, we treated organoids derived from Vil-creERT2 mice with 4-OH tamoxifen, using the standard protocol that we applied for in vitro deletion of ACC1. As shown in the figure below, we did not observe any effect of 4-OHT on organoid growth or formation of crypt domain structures, indicating that the phenotype we describe upon ACC1 deletion is not caused by a potential genotoxic effect or high concentration of tamoxifen. Regarding the potential in vivo genotoxic effect of tamoxifen, the publication by Bohin et al. (PMID: 30449703) reports a fast, but only short lived effect on stem cells after tamoxifen treatment (< 7 days), which is very different of what we observe in our system (phenotype develops > 5 days and is long lasting). It is also worth mentioning that addition of fatty acids can completely rescue the phenotype, which would be unlikely if genotoxic effects would contribute significantly to the phenotype. Finally, we confirmed

our phenotype independently of tamoxifen and Vil-CreERT2-mediated gene targeting using the specific ACC1 inhibitor Sorafenib. We believe that together, these findings speak clearly against a significant confounding effect of tamoxifen to explain the phenotype that we describe in our manuscript.

Figure legend: Crypts were isolated from Vcre^{ERT2}^{+/-} mice and grown in organoid cultures. 4-OHT (300 nM) was added to the organoids for 24 h at beginning of the culture. On day 5, organoids were imaged and the formation of crypt domains was quantified. >30 organoids were counted for each condition. Data was pooled from 2 independent experiments.

Q2: We would like more clarity on the model. Multiple lines of evidence suggest exogenous Wnt3a in culture should rescue the ACC1-KO phenotype: 1) beta-catenin is lowered in the mutant organoids and Wnt3a would stimulate more beta-catenin, 2) a HFD rescues the in vivo ACC1-KO histological phenotype in which the diet has been shown to increase Wnt signaling in ISCs. Can PPAR δ agonist rescue the in vivo phenotype? Or alternatively, does the loss of Ppard recapitulate this phenotype in vitro? The rescue experiment is very valuable here and your data are surprising. Regarding SuppFigD, does the addition of Wnt3a to control organoids have an effect, that is, is your batch of Wnt3a adequately functional. The results dealing with Wnt3a and ACC1-KO are unexpected and require additional considerations.

Author's response: We appreciate the essential concerns raised by the reviewer regarding the role of Wnt3a and PPAR δ in our system. We performed now additional experiments in order to define better the consequences of inducing Wnt signaling and the potential to rescue the ACC1-deficient phenotype. First, we independently confirmed that the Wnt3a we used for our studies is functional. To this end, we assessed the potential of Wnt3a to upregulate the expression of the Wnt-signaling target gene *AXIN2*, using a system with HELA cells (Doi. 10.1016/j.molcel.2014.04.014). We also cannot exclude that Wnt3a may not be the most effective activator of Wnt-signaling in our small intestine-derived organoid system, since the family of Wnt proteins contains several other members. In order to circumvent this potential limitation of using Wnt3a, we included the well-described compound CHIR99021 in our study, which acts more downstream of Wnt-signaling as a potent indirect activator of β -catenin. As shown in the figure below, both compounds, Wnt3a and CHIR99021, were able to upregulate *AXIN2* expression HELA cells, confirming their potential to induced Wnt/ β -catenin signaling. Yet, neither of the compounds rescued the effect of ACC1-deficiency in organoids (**new Supplemental Figure 4d**), confirming our previous results. These results indicate that de novo FAS (or the products of

de novo FAS) play a non-redundant function for ISC maintenance. As we have discussed in our manuscript, de novo FAS may not directly influence the Wnt/ β -catenin pathway but rather sustain Wnt signaling via indirect mechanisms. We agree with the reviewer that with regard to this aspect, it was necessary to further test the in vivo role of PPAR δ in our system. As suggested, we have now performed in vivo rescue experiments with the PPAR δ agonist GW501516 in ACC1 Δ/Δ IEC mice. Our results show that in vivo activation of PPAR δ prevented almost completely the histological signs of crypt loss and destruction in the small intestine on day 14 after tamoxifen treatment (**new Figure 4e**). Together with our in vitro data, this further proves that activation of PPAR δ is sufficient to compensate (at least to a significant extent) the defects caused by the loss of de novo FAS. As pointed out by the reviewer in question 3 below, Beyaz et al. (and more recently Mana et al.) demonstrated that genetic deletion of PPAR δ /PPAR α does not seem to have a critical impact on ISC function under normal physiological conditions. In their hands, loss of PPAR δ abrogated the ISC-stimulating effect of a high-fat diet. We show that PPAR δ activation can also be of functional relevance in a case where supply of fatty acids is blunted, such as upon loss of de novo FAS. It is therefore likely, that PPAR δ , and its ability to stabilize nuclear β -Catenin, might be most critical in situations of high- or missing fatty acid supply. To account for these considerations, we have adjusted the description of our findings and extended the discussion in the revised version of the manuscript.

Figure legend: HELA cells were treated with Wnt3a (100 ng/ml) or CHIR99021 (5 μ M) for 3h. qPCR was performed with primers specific for *AXIN2* and normalized to the expression of the housekeeping gene *HPRT1*. Data was pooled from 3 independent experiments.

Q3: In the Discussion (lines 349-351) “Our findings here support a model in which activation of PPAR δ is critical to sustain normal ISC function also under physiological conditions.” We find this statement to be an overinterpretation. Beyaz et al. use the organoid assay to show that Ppard loss had no effect on organoid growth, indicating it is not critical for ISC function under physiological conditions. Without further loss of function studies, this statement should be omitted or rephrased

Author’s response: We agree with the reviewer and apologize for overstating our findings. As already discussed in our response to question 2, we have rephrased our statements in the revised manuscript accordingly.

Minor.

-In organoid assays, it would be informative to know if the crypts were harvested from the proximal, distal, or total small intestine. Please indicate in the text or figure legends.

Author's response: The crypts that we used in our organoid cultures were harvested from total small intestines. We included this information to the revised version of our manuscript.

-Supp Fig1E. Please confirm the duodenum and ileum images are properly assigned. They appear swapped.

Author's response: We carefully checked the images that we used and confirmed their proper assignment.

-Supp Fig2 - Please clarify the label on the progenitor population "%Lgr5^{hi}"

Author's response: We thank the reviewer for pointing out this typo, which we have now corrected in the revised manuscript.

-Line 205. The sentence is unclear.

Author's response: We agree with the reviewer that this sentence was somewhat difficult to understand. We rephrased and shortened the sentence: "We used a conditioned growth medium, which promotes the growth of spherical organoids that lack differentiated cell types". The composition of this medium is described in detail in the methods part.

Reviewer #3 (Remarks to the Author):

On the positive side, the authors addressed the majority of the questions. On the negative side, some of the answers and conclusions are confusing.

Comments:

Q1. As mentioned by Reviewer 1, the phenotype of the ACC1 IEC KO mice does not consistently show the damage of crypt structure and appears to have limited impact on other cell types. For example, in Figure 1c, both proximal and distal colon show decreased goblet cells in the H&E image, as does the MMP7 staining in Figure 2d. However, in both Figures 2c and 2d, PAS staining shows no change in differentiated cell types. Moreover, in Sup Figure 1b, there appear to be more goblet cells in the ACC1 IEC KO colon. Perhaps there's an explanation. Bottomline this needs to be clarified.

Author's response: We appreciate the concerns raised by the reviewer regarding the consistency of our findings. We would like to point out that the phenotype of damaged or lost crypt structures was consistently observed in all ACC1^{Δ/ΔIEC} animals that we analyzed for this study by histology. We are also under the impression that the new images that we presented in Figure 1b and c of the revised manuscript do faithfully represent our finding of a gradual increase in the crypt pathology from the proximal to the distal parts of the small intestine and the colon. We agree with the reviewer that it can be difficult to assess the degree of goblet cell disturbance in the H&E stainings. In fact, we cannot rule

out the possibility that loss of de novo FAS may result in morphological changes in goblet cells, affecting their appearance in H&E stainings. Therefore, to unambiguously identify goblet cells and to allow for direct cell counting, we have performed PAS stainings, since we believe that this staining method gives a clear impression of goblet cells in histology. As shown in Figure 2d, careful assessment of PAS+ ileum goblet cell numbers on a per crypt basis did not reveal differences between ACC1^{Δ/ΔIEC} and control mice. As stated correctly by the reviewer, we did not observe a major impact on differentiated epithelial cell types following ACC1-deletion in vivo, except for a slight increase in the number of MMP7+ cells. Under in vivo conditions, the most significant changes upon ACC1 deletion were indeed the consistent drop in the expression levels of Lgr5 and a significantly reduced number of Olfm4+ ISCs. We have discussed the differences between our in vivo and in vitro findings in more detail in our reply to the reviewer's question 3 below.

Q2. In Figure 2e, how many crypts were isolated from Lgr5-EGFP WT and ACC cKO mice to do the sorting? For the flow map presented in Fig 2e, it's unclear how many crypts and cells were used which impacts statistical relevance.

Author's response: We are grateful to the reviewer for raising this issue. Thanks to the reviewer, we realized that the Y-axis labeling of Figure 2e was misleading. In fact, the percentages shown for ISCs were not normalized to crypt numbers, but reflect the percentages of Lgr5+ ISCs/progenitors in the total cell suspensions derived from the crypt isolation protocol. So, every data point in the figure represents the frequency of ISCs in the Duodenum/Jejunum and Ileum of one mouse. In total, we analyzed 7 mice per group from several independent experiments. We corrected the Y-axis labeling of Figure 2e and adjusted the description in the result and method parts of the revised manuscript. We hope that this clarifies the issue and apologize for the unclear labelling.

Q3. Following question 1, in Fig 3f, the RNAseq data of WT and ACC IEC KO mice also showed significant differences of marker genes of differentiated cell types. Do the authors have any explanation for the differences?

Author's response: We thank the reviewer for bringing up this important point. Indeed, the analysis of our RNA-Seq data revealed increased expression of markers associated with the secretory lineage or absorptive enterocytes in ACC1-deleted organoids. At the same time, we observed a strong downregulation of markers associated with stem cell identity and proliferation. The fact that ACC1-deletion results in a specific loss of stem cells may lead to a shift in the ratio between ISCs and other IEC types in the organoid cultures, which would explain (at least in part) the finding of increased transcripts of differentiated cell types in the absence of ACC1. In addition, we believe that several other factors may contribute to the differences that we observed between in vivo and in vitro conditions. One of these factors is the availability of external sources of fatty acids in vivo (but not in vitro), either through the standard diet or derived from the commensal microbiota. As we have explained in our previous replies to the other reviewers, we cannot exclude that escape from Vil-cre-mediated deletion in ISCs, as has been described by Berger et al. (Doi:10.1038/ncomms13171) might be more pronounced in vivo. We have included a discussion of these points in our manuscript (lines 402-412 in the discussion section). In addition, we also adjusted the text in the results part of the manuscript in order to better account for this fact.

4. Does GSEA of RNAseq data call out changes in fatty acid pathway genes?

Author's response: Our GSEA analysis showed that the most significantly affected pathways were associated with cell division and proliferation. We also observed pathways associated with the function of differentiated epithelial cell types, most prominently for antimicrobial and innate immunity-related functions. We believe that the latter is in accordance with the finding of increased transcripts for differentiated cell types in our RNA-seq. We included a list with the top 30 differentially expressed pathways to our revised manuscript (**new Supplementary Figure 3b**). However, we did not see an accumulation of fatty acid metabolism-related pathways in our analysis, which is in accordance with our finding that key enzymes for FAS or FAO were not differentially expressed between ACC1-deficient and control organoids (Figure 4a and supplementary Figure 5b).

5. In the AOM-DSS model, are tumors per colon representing one colon segment in each group or the average tumor numbers per mice? Please indicate the "n" number. Also please present more than one image per group. There also appear to be tumors in the ACC lox mice, but it is unclear whether they are not labeled or not counted or both. Also, because ACC1-deficient mice showed increased inflammation but reduction of tumors, I am unclear as to the authors' explanation "The finding that ACC1-deficient mice show rather enhanced inflammation-related pathology both in the acute phase of DSS treatment as well as at the endpoint of the experiment is of importance, since it demonstrates that reduction of tumors is not a simple consequence of reduced inflammation in those mice."

Author's response: We apologize for the lack of clarity. Every data point shown in Figure 6c represents the number of tumors counted in the complete colon of one mouse. In total, colons from n=19 (ACC1^{Δ/ΔIEC}) and N=13 (ACC^{lox/lox} mice) were analyzed from 3 independent experiments. We updated the Figure legend in the revised version of our manuscript to make this fact more clear. We are also grateful to the reviewer for this question, since we realized that the arrows we included in Figure 6b in the previous version of the manuscript were misleading, as they did not fully represent the numbers that were counted. As mentioned above, we counted all macroscopically visible tumors over the whole length of the colon. Although tumors were concentrated to the most distal part of the colon, we occasionally also observed tumors in the middle or proximal part. As we believe that the arrows did not provide any additional information, we omitted them in the updated version of the figure. Moreover, as requested by the reviewer, we now included the images of all colons from one of the 3 independent experiments into the **new Figure 6b**.

In the AOM/DSS model, tumor development upon AOM treatment is enhanced significantly by the DSS-mediated colonic inflammation. It is therefore important to rule out that less tumor development might simply be an indirect consequence of less inflammation. We believe that in this respect, it is of importance that ACC1^{Δ/ΔIEC} mice do not show less DSS-mediated inflammation, but rather enhanced pathology. Thus, the reduction in tumor numbers that we observed in ACC1^{Δ/ΔIEC} mice cannot be explained by reduced inflammation, but rather reflects the direct consequences of ACC1-deficiency on ISC biology. We agree that our explanation in the text was not very clear. We have rephrased our statement in the revised manuscript in order to make this point more comprehensible.

6. Although the authors explained the discrepancy of PPARdelta downstream genes (Cpt1a, Angptl4, Pdk1, Hmgcs2, etc) having no significant changes by proposing another role in promoting ISC function by stabilizing Beta-catenin function, we agree that further future studies are needed for delineating these two pathways.

Author's response: We thank the reviewer for the critical evaluation of our work and the valuable suggestions, which helped us a lot to increase the quality of our study. We believe that we were able to delineate important molecular mechanisms that are affected in ISCs upon loss of de novo FAS. Although we could not answer all remaining questions, we hope that the reviewer finds the present form of the manuscript suitable for publication.

REVIEWER COMMENTS

Reviewer #2 (Remarks to the Author):

The authors have adequately, clearly addressed many of my concerns. I support its publication.

Reviewer #3 (Remarks to the Author):

While the Authors' responses have been helpful, some in vivo and in vitro phenotypes are still not in alignment, in which the mechanism remains a bit confusing.

Comments:

1. In Figure 2e, is the sorting from ACC IEC KO mice or ACC Lgr5 KO mice? The label shows it is ACC KO in Lgr5 cells. The % Lgr5 cells needs to be labeled as % Lgr5 in total epi cell numbers. Also, how many epi cells were sorted from each mouse?

2. The in vivo and in vitro discrepancy of differentiated cells may affect the interpretation of the real mechanism of ACC deletion. Since the authors already have Lgr5-GFP mice, have you measured ACC's expression in Lgr5+ cells relative to Lgr5- cells? Also, have you done any immunostaining of differentiated cell gene markers such as Muc2 in ACC IEC KO organoids to verify the marked increased gene expression in Figure 3f? In theory, all these differentiated cells are differentiated from Lgr5+ cells. If ACC KO decreased ISC expression and organoid budding, where do the differentiated cell types come from? This leads to similar concern from the other reviewer that the increase of differentiated gene markers may be a side effect of tamoxifen in vitro use? Have you compared those ACC in vitro tamoxifen KO primary organoids with in vivo tamoxifen deletion?

REVIEWER COMMENTS

Reviewer #2 (Remarks to the Author):

The authors have adequately, clearly addressed many of my concerns. I support its publication.

Author's response: We would like to thank the reviewer for taking the time for a thorough and fair evaluation of our work. We are grateful for the valuable suggestions, which helped us a lot to increase the quality of our study.

Reviewer #3 (Remarks to the Author):

1. In Figure 2e, is the sorting from ACC IEC KO mice or ACC Lgr5 KO mice? The label shows it is ACC KO in Lgr5 cells. The % Lgr5 cells needs to be labeled as % Lgr5 in total epi cell numbers. Also, how many epi cells were sorted from each mouse?

Author's response: We would like to apologize for the lack of clarity in the description of the experiment. In fact, we did not sort cells for the experiment shown in Figure 2e, but performed direct ex vivo analysis. We compared frequencies of Lgr5+ ISC and progenitor cells in Lgr5-EGFP-IRES-CreERT2 reporter mice (which are ACC1 wild type) and Lgr5-specific ACC1-KO mice (Lgr5-EGFP-IRES-CreERT2 bred to ACC1lox/lox mice (-> ACC1Δ/ΔLgr5 mice). We have adjusted the description in the text of the revised manuscript to make this point more clear. We have also adjusted the description of the experiment in the Figure legend to indicate that we show frequencies of DAPI-Epcam+Lgr5high ISCs and DAPI-Epcam+Lgr5int progenitor cells within total DAPI-Epcam+ epithelial cells.

2. The in vivo and in vitro discrepancy of differentiated cells may affect the interpretation of the real mechanism of ACC deletion. Since the authors already have Lgr5-GFP mice, have you measured ACC's expression in Lgr5+ cells relative to Lgr5- cells? Also, have you done any immunostaining of differentiated cell gene markers such as Muc2 in ACC IEC KO organoids to verify the marked increased gene expression in Figure 3f? In theory, all these differentiated cells are differentiated from Lgr5+ cells. If ACC KO decreased ISC expression and organoid budding, where do the differentiated cell types come from? This leads to similar concern from the other reviewer that the increase of differentiated gene markers may be a side effect of tamoxifen in vitro use? Have you compared those ACC in vitro tamoxifen KO primary organoids with in vivo tamoxifen deletion?

Author's response: We appreciate the concern of the reviewer regarding some of the discrepancies that we observe in our in vivo and in vitro experiments. Following the reviewer's advice, we have directly measured the expression of ACC1 in Lgr5+ ISCs sorted from the intestine of Lgr5-EGFP reporter mice. As shown in the **new Supplementary Figure 2a**, we find that both Lgr5+ and Lgr5- epithelial cells express comparable levels of ACC1. The expression level is in the range of what we have observed for other cell types, e.g. T cells (doi:10.1038/nm.3704). Although this data confirms that Lgr5+ ISC do express ACC1, we would like to mention that regulation of de novo FAS seems to be rather not on the level of transcription. As we have described in our answers to reviewer 1 during the first revision of this manuscript, this conclusion is not only supported by our own data (Figure 4a), but also by previous studies which showed that interference with de novo FAS did not result in major changes in the transcription-levels of de novo FAS-associated genes (doi:10.1038/nature11689). This is also in accordance with the idea that ACC1 activity is mainly regulated on the post-transcriptional level, e.g. by phosphorylation-induced multimerization or allosteric activation or inhibition (Doi.org/10.1016/j.it.2014.12.005).

Following further the suggestions of the reviewer, we have now performed direct staining for Muc2 in organoids (**new Supplemental Figure 3a**). Our results indicate that there is indeed an increase in the number of Muc2 positive cells upon deletion of ACC1, confirming the results of our RNA-seq. As stated correctly by the reviewer, all differentiated cell types are supposed to differentiate from stem cells. In our organoid system, ACC1 is deleted at the early spheroid stage that consists mainly of stem cells. We think that that loss of ACC1 tips the balance of stem cell maintenance by proliferation towards differentiation of ISC into IEC lineages. This is consistent with the strong downregulation of markers associated with stem cell identity and increased expression of markers associated with the secretory lineage (Figures 3e, f). This would also explain why we see less budding and reduced capacity of ACC1-deleted organoids in secondary cloning experiments (Figure 3 c, h). We appreciate the concerns regarding potential side effects of tamoxifen in our system. Yet, we believe that there are several lines of evidence speaking against such a side effect. As we have described in our previous answer to reviewer 2, we have shown that organoids derived from Vil-creERT2 mice do behave similar to non-treated organoids and do not show any defect upon tamoxifen treatment. In addition, we demonstrated that direct inhibition of ACC1 by a pharmacological inhibitor mirrors the effects of ACC1 deletion independent of tamoxifen treatment (Figure 3k). Although we cannot formally rule out that tamoxifen treatment may result in upregulation of differentiated gene marker as a direct side effect, we believe that this is very unlikely. In particular, when taking into account that tamoxifen is withdrawn from the organoids after 24h of treatment and upregulation of differentiated genes was determined 4 days later (Figure 3 e,f). Independent of these considerations, we think that our data using ACC1^{Δ/ΔLgr5} mice (Figure 2e) and direct inhibition of ACC1 in Lgr5+ ISC-derived organoids (Figure 3k) together clearly demonstrate that ACC1 is of specific importance for ISCs both under in vivo and in vitro conditions. We still believe that the minor differences that we observed between in vitro and in vivo could be best explained by the environmental conditions, such as the presence of fatty acids or ISC that escape ACC1 deletion in vivo, as we have discussed in our manuscript.

REVIEWERS' COMMENTS

Reviewer #3 (Remarks to the Author):

The authors have adequately addressed my concerns, and I support publication.

REVIEWER COMMENTS

Reviewer #3 (Remarks to the Author):

The authors have adequately addressed my concerns, and I support publication.

Author's response: We would like to thank the reviewer for taking the time for the evaluation of our work. We are grateful for the valuable suggestions, which helped us a lot to increase the quality of our study.